# Continuous online-monitoring of Ice Nucleating Particles: development of the automated Horizontal Ice Nucleation Chamber (HINC-Auto)

Cyril Brunner[1] and Zamin A. Kanji[1]

[1]Institute for Atmospheric and Climate Science, ETH, Zurich, 8092, Switzerland

**Correspondence:** C. Brunner (cyril.brunner@env.ethz.ch) and Z.A. Kanji (zamin.kanji@env.ethz.ch)

**Abstract.** The incomplete understanding of aerosol-cloud interactions introduces large uncertainties when simulating the cloud radiative forcing in climate models. The physical and optical properties of a cloud, as well as the evolution of precipitation, are strong functions of the cloud hydrometeor phase. Aerosol particles support the phase transition of water in the atmosphere from a meta-stable to a thermodynamically preferred, stable phase. In the troposphere, the transition of liquid droplets to ice crystals in clouds, via ice nucleating particles (INPs) which make up only a tiny fraction of all tropospheric aerosol, is of particular relevance. For accurate cloud modeling in climate projections, the parameterization of cloud processes and information such as the concentrations of atmospheric INPs are needed. Presently, only few continuous real-time INP counters are available and the data acquisition often still requires a human operator. To address this restriction, we developed HINC-Auto, a fully automated online INP counter, by adapting an existing custom-built instrument, the Horizontal Ice Nucleation Chamber. HINC-Auto was able to autonomously sample INPs in the immersion mode at a temperature of 243 K and a water saturation ratio of 1.04 for 97% of the time for 90 consecutive days. Here we present the technical setup used to acquire automation, discuss improvements to the experimental precision and sampling time, and validate the instrument performance. In the future, the chamber will allow a detailed temporal analysis (including seasonal and annual variability) of ambient INP concentrations observing repeated meteorological phenomena compared to previous episodic events detected during campaigns. In addition, by deploying multiple chambers at different locations, a spatio-temporal variability of INPs at any sampling site used for monitoring INP analysis can be achieved for temperatures $\leq$ 243 K.

## 1 Introduction

The interaction between aerosols and clouds contributes to the global energy budget by indirectly influencing the radiative forcing of the climate system. Yet, predictive climate models struggle to accurately simulate aerosol-cloud interactions, e.g. the Intergovernmental Panel on Climate Change attributed a low confidence level of the aerosol-cloud interactions in their fifth assessment report (Boucher et al., 2013). Clouds containing ice have special relevance to the Earth's climate. Not only does the cloud phase strongly influence climate-relevant physical properties, such as albedo (e.g., Sun and Shine, 1994; Lohmann and Feichter, 2005), but ice is also shown to be responsible for the development of most mid-latitude precipitation processes (e.g., Mülmenstädt et al., 2015). Consequently, it is essential to understand the mechanism of ice formation within mixed-phase

clouds. One such mechanism is immersion freezing, the formation of ice crystals on ice nucleating particles (INPs) immersed in liquid droplets (Vali et al., 2015), which has been studied extensively in laboratory studies (see e.g., Zuberi et al., 2002; DeMott et al., 2003; Marcolli et al., 2007; Lüönd et al., 2010; Niemand et al., 2012; Murray et al., 2012; Atkinson et al., 2013; Hiranuma et al., 2015). A large number of field studies based on intensive observation periods to quantify the concentration, properties, identity and sources of immersion mode INPs have also been reported (see e.g. Dufour, 1862; Rogers and Vali, 1978; Demott et al., 2003; Richardson et al., 2007; Chou et al., 2011; Boose et al., 2016; Lacher et al., 2017). These studies have substantially improved our understanding of such atmospheric INP properties and sources, but some aspects such as diurnal variability, seasonal and and annual trends as well as general spatio-temporal variability of INPs remains poorly constrained (see e.g., Demott et al., 2011; Cziczo et al., 2017; Kanji et al., 2017; Lacher et al., 2018) and insufficiently understood, needing more research to accurately represent atmospheric ice formation in climate models (Demott et al., 2010; Phillips et al., 2013). For example most field studies on INP measurements in the atmosphere have to focus on cases rather than drawing conclusions about the long term trends because of the lack of such data sets. To quantify the free troposphere INP concentration in the Swiss Alps, data from 9 field campaigns over the course of 3 years was required in order to quantify this parameter representing a background INP concentration at a single temperature (242 K) (Lacher et al., 2018), making such research data costly and inaccessible to cloud modellers. Furthermore, with the lack of long term frequent data sets, cloud model research is limited to the small specific case studies for validating INP parameterizations in global or regional climate models (see e.g., Niemand et al., 2012).

To sample INPs, either offline or online techniques are available. In offline INP counters, air is sampled through a filter or an aerosol-to-liquid cyclone impinger. The filter is washed in pure water to extract the sampled aerosols while the cyclone is filled with a small amount of water in which the aerosols accumulate. The sample is then divided into small droplets on cold stages. The freezing of the droplets as a function of temperature allows deducing the INP concentration of the sampled air (Vali, 1971). The advantages of the offline technique are the ability to detect low INP concentrations thus reporting INP concentrations at fairly low (272 K) as well as high supercooling (235 K) (e.g., Conen et al., 2015; Petters and Wright, 2015; Mignani et al., 2019; Wex et al., 2019; Brubaker et al., 2020). Furthermore, the ability of having the aerosols contained for future additional analysis is also possible because not all the sample has to be used in this method of processing and neither is the analysis method of the sample destructive. This comes at the sacrifice of lower time resolution due to the continuous sampling for 8 - 12 hours or more per sample, in order to generate appropriate signal to noise ratio in the freezing spectra. Additional drawbacks include the chance for sample modification when INPs impact the filter or during sample handling and storage (e.g., Cziczo et al., 2017, and references therein). Most recently, Beall et al. (2020) found losses of up to 72% to INP concentration caused by storage at room temperature versus losses of 25% by storage at 253 K over up to 166 days.

Online techniques sample and detect INPs in one step in real time. The measurement of INPs in ambient air is challenging since their number concentrations are on the order of $10^{-1}$ to $10^2$ std $L^{-1}$ (at 242 K, $S_w$ = 1.04, (see e.g., Lacher et al., 2018; Kanji et al., 2017)) while the sensitivity of portable INP counters is on the order of $10^{-2}$ std L $^{-1}$ (Cziczo et al., 2017). This

makes online counters good candidates for reporting INP concentrations at moderate (with the presence of aerosol concentrators) to high supercooling ($\leq$ 248 K). Online counters have the advantage of resampling INPs downstream of cloud chambers for further single particle analysis, but often times these measurements are technically challenging and time consuming because of the low concentrations of INPs. Online and offline data acquisition of INP concentrations often needs human operation and is subsequently limited to isolated or planned field campaigns. A prime limitation for the absence of long term monitoring data sets was that online real-time measurements of INP concentrations via INP counters required human operators as no autonomous online INP counter were available. Bi et al. (2019) presented the first autonomous online INP counter based on a CFDC. A novel paper by Möhler et al. (2020) introduced the Portable Ice Nucleation Experiment (PINE), an autonomous online INP counter that uses the adiabatic cooling during expansion to activate the INPs at the targeted supersaturation. Offline techniques can be automated in their sampling but require substantial sample handling, storage, and finally processing of samples to derive INP concentrations, all by dedicated trained scientists. Furthermore automated aerosol sampling techniques will suffer significantly lower time resolution with the advantage of quantifying INP concentrations at higher temperatures compared to online techniques (Cziczo et al., 2017, and discussion therein).

In this work we present an automated continuous online INP counter, HINC-Auto. Besides a technical description and findings to enhance the chamber precision and sampling time, the chamber is validated with laboratory tests and experiments are compared to a theoretical model. HINC-Auto has been implemented for continuous monitoring of INPs at center temperature ($T$) = 243 K and and saturation with respect to water ($S_\mathrm{w}$) = 1.04 at the High Altitude Research Station Jungfraujoch (JFJ, 3580 m a.s.l., 46°33' N, 7°59' E). From the collected data set, since Feb 2020, it is expected that a continuous and more detailed temporal analysis of ambient INP concentrations will be possible, thereby observing repeated meteorological episodes compared to previous singular events detected during a single field campaign.

## 2 Materials and Methods

### 2.1 Working principle

In the 1980s, continuous flow thermal gradient diffusion chambers were developed to study cloud condensation and ice nucleation (Hussain and Saunders, 1984; Al-Naimi and Saunders, 1985; Tomlinson and Fukuta, 1985; Rogers, 1988). Continuous flow chambers (CFC) used a temperature gradient $\nabla T$ to produce water vapor supersaturation in the region between two water- or ice-covered walls. Rogers (1988) defined the term continuous flow thermal gradient diffusion chamber (CFDC) and described the underlying working principle: The annular volume between two vertically oriented concentric cylinders forms the chamber's cavity. The outer and inner walls of the chamber are chilled individually and are covered with a thin layer of ice. For the experiment, one wall is held at a warmer temperature than the other wall, thereby, heat as well as water vapor are diffused from the warm to the cold wall, forming radially linear steady-state temperature and water vapor pressure fields. Because the saturation vapor pressure with respect to water and ice are exponential functions, a supersaturation with respect to ice, $S_\mathrm{i} > 1$, is formed between the two walls. Supersaturation with respect to water $S_\mathrm{w} \geq 1$ can be achieved with large enough

temperature gradients. A steady flow passes coaxially through the chamber, entraining the sample aerosols in its center lamina. The surrounding sheath air is dried and filtered before entering the chamber. $T$ and $S$ are adjusted to the desired experimental conditions. A more detailed description can be found in Rogers (1988).

The CFDC of Kumar et al. (2003) was developed to probe aerosols and investigate their ability to acts as cloud condensation nuclei (CCN). In contrast to the design of Rogers, the chamber consisted of two horizontally oriented parallel walls. Kanji and Abbatt (2009) altered the design to the University of Toronto CFDC (UT-CFDC) to study ice nucleation at low temperatures. A horizontally oriented ice nucleation chamber offers two major advantages: first, the buoyancy of the air within the chamber resulting from the temperature gradient stabilizes the desired laminar flow. In vertically oriented chambers, the flow is oriented from the top to the bottom. The buoyancy at the warm wall favours an upward vertical air movement which counteracts the overall flow direction. Likewise, the reduced buoyancy at the cold wall adds a relative sinking motion in the proximity of the cold wall. If the temperature gradient is increased such that the shear between the lamina overcomes the fluid's viscous forces, the flow becomes turbulent. The increased mixing not only dislocates the sample particles away from their desired center lamina but also equalizes the temperature and water vapor profiles. The set conditions then may deviate from the assumed values (Rogers, 1988; Stetzer et al., 2008). This happens at gradients larger than 10-15 K, depending on the center lamina temperature (Garimella et al., 2016). Secondly, in horizontally oriented chambers the vapour diffused to the cold wall forming frost tends to stay on the bottom wall because of gravity. In a vertical ice nucleation chamber, the ice grown on the cold wall has the tendency to break off, get carried along with the downward moving sheath flow and is in danger of being misinterpreted as ice formed on an INP. The biggest drawback of the horizontal orientation is the sedimentation of hydrometeors or large aerosol particles. The hydrometeors have to grow to a size of $d \geq 1\,\mu\mathrm{m}$ to be detected by the sensor placed at the outlet. For particles $\geq 5\,\mu\mathrm{m}$, the terminal sedimentation velocity is such that the hydrometeors are subject to gravitational settling and, depending on their residence time ($\tau$), will not be sampled by the detector.

Lacher et al. (2017) presented the Horizontal Ice Nucleation Chamber (HINC), a close adaption of the UT-CFDC. HINC's record sampling times of 14 h make the chamber's design best suited to be adopted for continuous measurement of ambient INPs. A basic illustration of HINC is shown in Figure 1.

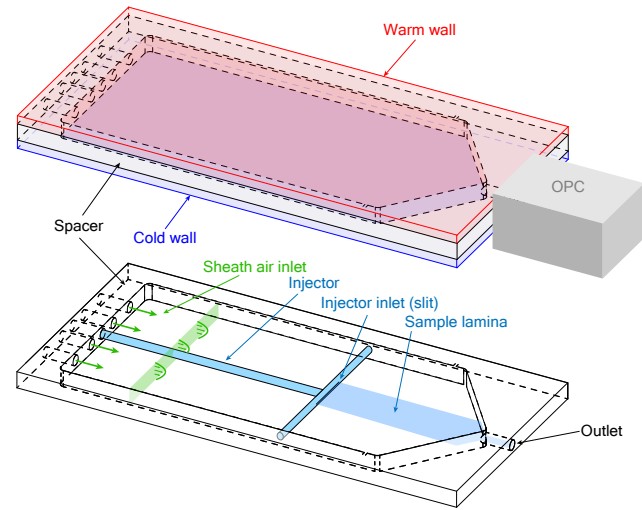

**Figure 1.** A schematic of HINC, the building platform of HINC-Auto. Top: the entire chamber, bottom: the internal parts.

## 2.2 HINC-Auto technical setup

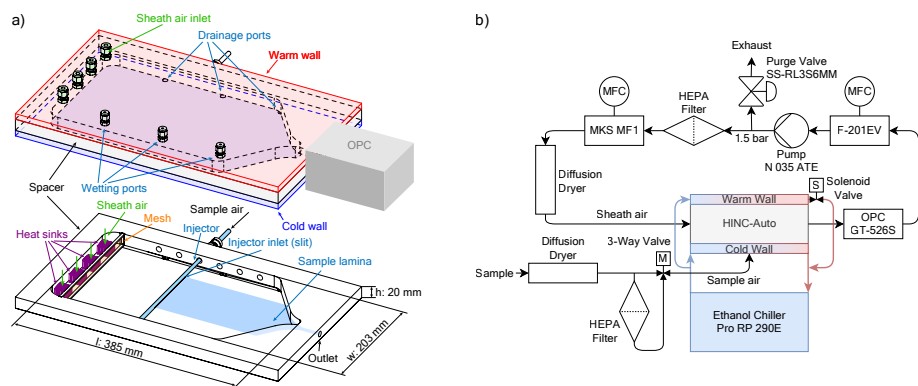

**Figure 2.** A schematic of HINC-Auto. a) Top: the entire chamber, bottom: the internal parts. b) External components and flow setup.

Figure 2a shows a schematic of HINC-Auto and 2b the external components and flow setup. The chamber consists of two aluminum cooling walls, with a 25 µm copper plating to avoid growth of mold. The chamber walls are cooled with an external recirculating ethanol chiller (Lauda PRO RP 290E), operated in a parallel-flow constellation to the air flow within the chamber. A polyvinylidene fluoride (PVDF) spacer physically and thermally separates the two chamber walls. A cut-out section within the PVDF spacer and the inner surfaces of the metal walls forms the cavity of the chamber. The inner metal walls are each covered with one layer of self-adhering borosilicate glass microfibre filter paper (PALL 66217, 1 µm, 8x10") which is wetted with water and acts as reservoir for water vapor in order to create ice and/or water (super)saturation. During normal operation, the temperature of the bottom wall is equal to or lower than the one of the top wall. The chillers PID-controller directly controls

the bottom wall temperature. A Y-connection just upstream of the bottom wall cooling inlet allows some of the chilled ethanol flow to diverge and flow through the top wall. At the top wall cooling exit, a solenoid valve controls the flow rate of the ethanol through the top wall before feeding it back into the re-circulating bath via the ethanol outlet of the bottom (colder) wall. If the top (warmer) wall is too warm the solenoid is actuated by a PI-controller and if the top wall is too cold, a 12V fan blows room air at the outer surface of the warm wall to increase its temperature. Three thermocouples (Transmetra TEMI313, type K, NiCr-Ni) monitor the temperature on each wall. To expose the aerosol particles within the sample air to a predefined temperature and supersaturation, the sample air has to remain in the center plane between the two chamber walls. Therefore, the center lamina is sandwiched between equal parts of particle-free sheath air. The sheath to sample air ratio is 9 to 1. The mass flow controlled (mass flow controller (MFC) MKS MF1, full scale range: 5 std L min$^{-1}$, set to 2.547 std L min$^{-1}$) and dried (diffusion dryer, $S_w \leq 0.008$ at 20 °C, filled with 4 Å-molecular sieve) sheath air enters the chamber through four holes in the top wall and is blown on to copper heat sinks mounted on the bottom wall. The sheath air is thereby rapidly chilled and flows through a mesh which equilibrates the flow and decreases the degree of turbulence. The weaving plain type 304-stainless steel wire mesh has a mesh-size of 250 mesh inch$^{-1}$ and a wire diameter of 0.04 mm. It spans over the entire width and height of the chamber's cavity (See Fig. 2a). The filter paper coats the cooling walls only downstream of the mesh. A horizontally aligned injector with a outer diameter of 6.35 mm and a slit of 0.4x100 mm$^2$ is used to guide the sample air into the chamber. The sampled air also passes a diffusion dryer identical to the sheath flow dryer before entering the injector. The injector is mounted through one of six holders in the side wall of the spacer. All unused holes are plugged. Placing the injector in a hole further upstream allows to increase the particle residence time and vice versa. A 6-channel optical particle counter (OPC, MetOne GT-526S) detects the number and size of the particles exiting the chamber via the outlet. A MFC downstream of the OPC (Bronkhorst, F-201EV, full scale range: 3.5 std L min$^{-1}$) is set to 2.83 std L min$^{-1}$ (defined by the specifications of the OPC). The sample air flow rate of 0.283 std L min$^{-1}$ results from the difference of the volume flow exiting the chamber through the OPC and the sheath air directed into the chamber. Consequently, a well sealed chamber is crucial for a representative operation. Downstream of the OPC and the MFC, a vacuum pump (KNF, N 035 ATE) is used to generate sufficient pressure drop over the MFC and to increase the pressure after the pump. A purge valve (Swagelok, SS-RL3S6MM) purges excess air above the set absolute pressure of 1.5 bar. The air is filtered with a HEPA filter and recycled back into the sheath air MFC.

During operation, water vapor diffuses from the top to the bottom wall, depleting the top wall of the ice layer while adding additional ice to the bottom wall. To maintain the desired supersaturation within the chamber, the top wall ice layer needs to get replenished by re-wetting the filter paper at temperatures $T > 273$ K. This rewetting procedure requires tilting the chamber by 25° using a linear motor to allow excess water and water collected on the cold wall to drain. A peristaltic pump with a flow rate of 10 ml min$^{-1}$ pumps 40 ml water from a water reservoir into one of the three wetting ports, located on the top cooling plate. Two solenoid valves alternate the used wetting port every 10 seconds. During the rewetting procedure a second peristaltic pump with a flow rate of 65 ml min$^{-1}$ is used to drain excess water which is recycled back into the rewetting reservoir. The reservoir is initially filled with 500 ml double-deionized water. The reservoir is protected from day light while a piece of copper and a UV-LED protects the water from spoilage. The total water uptake of the chamber is 100 ml month$^{-1}$. The molecular

sieves in the diffusion dryers need to be replaced every 30-60 days, depending on the ambient relative humidity. Sensors are used to check both sheath and sample air relative humidity to determine when the molecular sieves require replenishing.

### 165  2.3  Design changes

HINC showed differences between the measured and the calculated particle residence times as shown in Figure 3a. The particle residence times were measured with pulse experiments, where a brief pulse of particles was injected into HINC and then compared to the delayed temporal evolution of the particle counts exiting the chamber. To analyze the discrepancy, a 3D computational fluid dynamics (CFD) simulation was carried out (STAR-CCM+ V13.04.010). The CFD simulation was validated
with a particle image velocimetry (PIV) experiment. A detailed description of CFD simulation and the PIV experiment can be found in the appendix (A1). Only after modelling the tubing of the sheath air and corresponding union tee and union cross to distribute the sheath air outside of the chamber, the CFD simulation and the PIV-experiment came into good agreement. The validated simulation showed persistent high velocity regions downstream of the sheath air inlets (see Figure A2) which are not in agreement with the anticipated flow velocities. This explains the lower residence time in pulse tests compared to the cal-
culated residence time, which assumed ideal velocity distributions within HINC. Besides the high velocity jet regions (Figure A2), reversed flows are also present within the chamber, forming two large counter rotating vortices. Particle sedimentation experiments with HINC also confirmed the two vortices. To smooth the flow field within the chamber and achieve a more consistent desired unidirectional flow field, a mesh was introduced 20 mm downstream of the sheath air injector holes. PIV experiments with the mesh installed in HINC showed the anticipated homogenization of the flow velocity. With HINC-Auto
pulse tests are now in good agreement with the calculated values as seen in Figure 3b. Therefore, no additional PIV experiments and CFD simulations were performed with HINC-Auto.

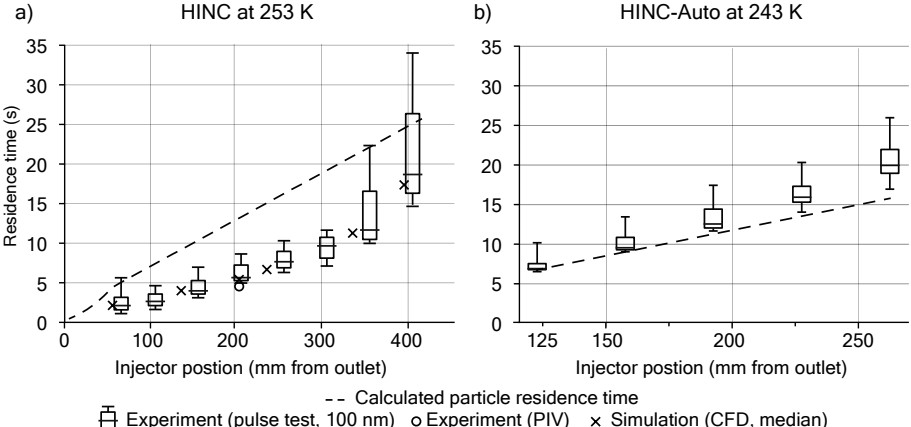

**Figure 3.** Calculated and measured particle residence time in a) HINC (without mesh) for different injector positions at $T$ = 253 K and b) HINC-Auto (using a mesh to achieve a more uniform flow) at $T$ = 243 K. Box plots from pulse experiments: median with 25/75% quartiles, whiskers: 5/95% quantiles. Median of PIV experiment (circles, $T$ = 288 K) and CFD simulation (crosses, $T$ = 243 K).

### 2.3.1 Reduction of rewetting duration

Maximizing the time HINC-Auto is sampling ambient air for INPs requires the duration of the rewetting procedure to be minimal. The limiting factor is warming up and cooling down the chamber from the working temperature (i.e. 243 K) to temperatures above the melting point of water (e.g. 293 K) and vice versa. The time to warm up and cool down the chamber is mostly set by the chiller's performance and by the heat capacity of the chamber itself. The new chiller (Lauda PRO RP 290E) had the best performance in comparison to the available products and their parameters, such as cooling power between 290 and 240 K, size, weight and price. Furthermore we decided to minimize the chamber's total heat capacity by machining the walls in aluminum instead of copper. A heat conduction analysis using the Finite Element Method in STAR-CCM+ V13.04.010 revealed sufficient temperature equality $\leq 0.02$ K at $T = 243$ K across both walls when changing from copper to aluminum, despite the 70% lower thermal conductivity. The 3D CFD simulation revealed for the supersaturation to need a substantial part of the chamber length to equilibrium to the set conditions. For $T = 243$ K and $S_w = 1.04$ a 96% equilibration ($S_w = 1.00$) was reached just 20 cm of the sheath air inlets (43% of the entire chamber length in HINC). A faster equilibration allows for a reduction of the chamber length, and subsequently, a further reduction in total heat capacity. To be more time efficient in studying the impact of the chamber length, a numerical 2D diffusion model approach was chosen over 3D CFD simulation. The newly developed 2D diffusion model has been validated with the analytical solution by Rogers (1988) and the 3D CFD simulation (see Figure A4). In order for the supersaturation to equilibrate to the set supersaturation, the temperature as well as the water vapor distribution along the chamber's height have to equilibrate from the initial conditions. Figure 4a shows the equilibration of the supersaturation profile in HINC to be temperature limited when injecting sheath air at $T_{sheath\ air} = 298$ K with a dew point of $T_d = 233$ K while the chamber is set to $T = 243$ K and $S_w = 1.04$. By precooling the sheath air to $T_{sheath\ air} = 248$ K the chamber can be shortened by 10 cm to facilitate an identical degree of supersaturation equilibration during the final 10 seconds (see Figure 4b). Larger residence times than 10 seconds are prone to hydrometeor sedimentation and are therefore not targeted. Precooling is achieved by blowing the sheath air onto heat sinks, mounted on the cold wall just upstream of the mesh. The mentioned changes and the new chiller decrease the duration of rewetting from 110 min with HINC to 50 min with HINC-Auto.

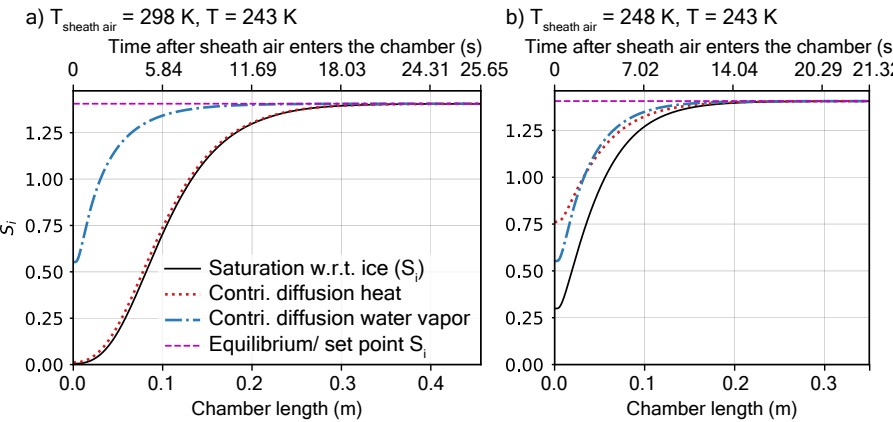

**Figure 4.** Simulated ice supersaturation development along the chamber's center lamina at $T$ = 243 K and contributing factors diffusion of heat and water vapor for a) $T_{sheath\ air}$ = 298 K (original length) and b) $T_{sheath\ air}$ = 248 K (allowing for the chamber length to be reduced by 10 cm).

### 2.3.2 Change of the axis of rotation during the rewetting procedure

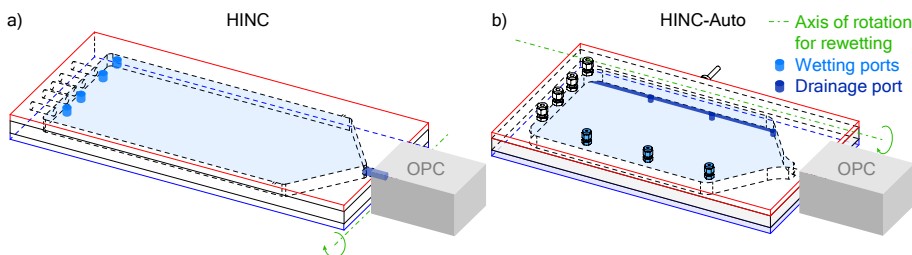

**Figure 5.** Illustration of the axis of rotation, location of wetting and drainage ports on a) HINC and b) HINC-Auto.

When rewetting HINC, the OPC has to be disconnected from the chamber's outlet to allow excess water to drain. HINC is thereby tilted 23° around an axis parallel to the width of the chamber (see Figure 5a). During rewetting HINC-Auto is tilted 25° around an axis parallel to the length of the chamber. This allows for the OPC to stay attached but required 3 additional
210 drainage ports in the bottom wall, located just below the spacer's side wall(see Figure 5b).

### 2.3.3 Software

The chamber is controlled via a newly developed guided user interface programmed using Python 3.7 and corresponding open source packages. The postprocessing of INP concentrations is done in real time. HINC-Auto can be accessed and controlled remotely if an internet connection is available on site, however this is not a requirement for autonomous operation. A screenshot
of the guided user interface with comments on the basic parameters a user can set is shown in the appendix (A7 -A11).

## 2.4 Derivation of the INP concentration

Immersion mode INP measurements are performed by sampling at $T = 243$ K and $S_w = 1.04$. At these conditions, CCN should activate to supercooled droplets, and INPs, which are active at 243 K, form ice crystals. The size of the hydrometeors exiting the chamber is used to differentiate between liquid droplets and ice crystals because ice grows to larger sizes at the prevailing conditions ($S_i \gg S_w$). Differential measurements between the total aerosol inlet and immediately downstream of the chamber with an OPC were used to determine particle losses. The transmission fraction of ambient particles $\geq 2$ μm through the tubing and the dry chamber (both walls held at 293 K) on the JFJ is 33%. No ambient particles $\geq 3$ μm were transmitted. Therefore, to assess the maximum size of droplets in the following diffusional growth calculations a maximum initial radius of $d_0 = 2$ μm is used.

Diffusional growth calculations (Rogers and Yau, 1989) with set fixed $T$ (e.g. constant at 243 K) and $S_w$ (e.g. constant at 1.04) conditions overestimate the final hydrometeor size at the chamber exit since the calculation assumes a constant supersaturation to be maintained for the entire time the particle passes through the chamber. In reality, the saturation in the particle stream needs to equilibrate to the set conditions, thus, the particles are exposed to a lower saturation for the first few seconds (see Figure A4). The 2D diffusion model provides an estimate of the real $T$ and $S_w$ when using the diffusional growth calculations by Rogers and Yau (1989). For an initial diameter of $d_0 = 2$ μm, liquid droplets are calculated to grow to a maximum size of $d_{liq} = 3.31$ μm (Zurich, 965 hPa, $\tau = 9.1$ s) and $d_{liq} = 2.36$ μm (JFJ, 645 hPa, $\tau = 6.1$ s). Measurements of a highly hygroscopic aerosol, ammonium nitrate with an initial mobility diameter of $d_m = 200$ nm (for the sample preparation see Section 3.2) show the onset of cloud droplets (no ice crystals since $T > 235$ K) in the $\geq 3$ μm-size bin at $S_w = 1.038$, as seen in Figure 6a, and support the calculated maximum size of 3.31 μm at $S_w = 1.04$. The impact on the final diameter for an initial size of $d_{0_{0.2}} = 200$ nm compared to $d_{0_{2.0}} = 2$ μm is 0.63 μm ($d_{liq_{2.0}} = 3.31$ μm vs. $d_{liq_{0.2}} = 2.68$ μm at 965 hPa and $\tau = 9.1$ s). If the INPs activate as soon as ice saturation is exceeded, the ice crystals grow to $d_{ice_{2.0}} = 7.77$ μm, $d_{ice_{0.2}} = 7.51$ μm at 965 hPa and $\tau = 9.1$ s and $d_{ice_{2.0}} = 7.66$ $\mu m$ (JFJ, 645 hPa, $\tau = 6.1$ s). Therefore, for experiments performed at $T = 243$ K and $S_w = 1.04$, all particles detected in the size bin $\geq 4$ μm are considered to be ice crystals formed on INPs. Figure 6b shows a measured activated fraction ($AF$) curve of ambient air on the JFJ during a high INP concentration period (7:05 22. March 2020, UTC). $AF$ is the ratio of all particles, that are detected in the indicated size bin, to all sampled particles, measured with a CPC within the sample flow. The onset of cloud droplets in the $\geq 0.3$ μm size bin exactly at $S_w = 1$ demonstrates the accuracy of HINC-Auto. At $S_w = 1.13$ an observed steep increase in $AF$ in the $\geq 3$ μm-OPC size bin indicates droplets only grew larger than 3 μm at this $S_w$. Compared to the ammonium nitrate measurements performed at Zurich, a delayed activation is observed. This is expected because of the decrease in ambient pressure, which results in shorter residence times, and the much lower hygroscopicity of ambient particles at the JFJ compared to ammonium nitrate. The signal visible in the $\geq 4$ μm-OPC size bin comes from INPs, which nucleate and grow to ice crystals at $S_w \geq 1.028$ ($S_i \geq 1.378$). This validates the calculations above that at $S_w = 1.04$ droplets cannot grow to sizes $\geq 4$ μm but ice crystals can supporting the use of the $\geq 4$ μm size bin to detect ice crystals.

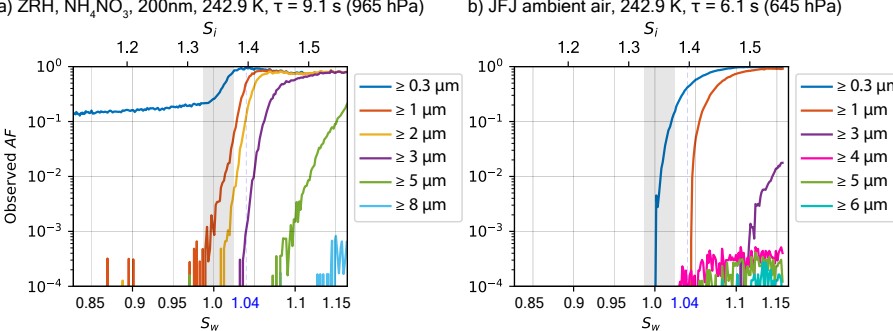

**Figure 6.** Activation curve at $T$ = 243 K for a) ammonium nitrate, sampled at Zurich and b) ambient air, sampled at JFJ during an period with enhanced INP concentrations. Both measurements were performed with HINC-Auto with an identical injector position, but resulted in a shorter particle residence time ($\tau$) at the JFJ compared to Zurich because of the reduced ambient pressure. Grey shading refers to chamber uncertainty (see Section 3.1 for details).

False positive counts can arise from large particles other than ice nucleated on an INP. Dominant false positives arise from frost grown on inner chamber surfaces which break off and get carried with the prevailing airflow until exiting the chamber, where they are detected by the OPC. To assess and correct the measurements for these false counts, before and after a sampling period of 15 min a background measurement of 5 min is performed. During the background measurement the sample air is directed through a HEPA filter before being sampled in the chamber. The mean time-normalized background counts before

and after each INP measurement in the $\geq 4$ μm bin are subtracted from the $\geq 4$ μm-OPC counts during the INP measurement before the conversion to std L min$^{-1}$. The INP concentration is calculated as follows:

$$INP\ concentration = \left( \frac{\sum INP\ counts}{\sum N_{INP\ samples}} - \frac{\sum BG\ counts}{\sum N_{BG\ samples}} \right) \frac{1}{Vt_{OPC}} \tag{1}$$

where $\sum INP\ counts$ is the sum of all counts (particle number) in the $\geq 4$ μm OPC size bin during the INP measurement, $\sum N_{INP\ samples}$ is the total number of OPC intervals during the INP measurement, $\sum BG\ counts$ is the sum of the back-

ground counts (particle number) in the $\geq 4$ μm OPC size bin before and after the INP measurement while sampling through a particle filter, $\sum N_{BG\ samples}$ is the total number of background OPC intervals before and after the INP measurement, $t_{OPC}$ is the duration of each OPC interval in minutes (here 5 sec, thus 5/60 min), and $V$ is the sample flow rate, here $V$ = 0.283 std L min$^{-1}$. As the volume flow through the OPC is controlled by the MFC in std L min$^{-1}$, the resulting INP concentration is INP std L$^{-1}$.

The limit of detection (LOD) is calculated as follows:

$$LOD = \frac{\sqrt{\sum BG\ counts}}{\sum N_{BG\ samples}} \frac{1}{Vt_{OPC}} \tag{2}$$

where the LOD is in std L$^{-1}$. If over a period of 120 OPC background sampling intervals with a duration of 5 seconds each a total of 3 counts where detected in the $\geq 4$ μm OPC size bin, the LOD would be = 0.612 std L min$^{-1}$ ($\sum N_{BG\ samples}$ = 120, $t_{OPC}$ = 0.083 min, $\sum BG\ counts$ = 3, $V$ = 0.283 std L min$^{-1}$). The stated LOD provides a 62.3% (1 $\sigma$) confidence

interval. The minimum detectable concentration (MDC) is 1 count (particle) in the $\geq 4\ \mu m$ OPC size bin over a 15-minute INP measurement with a sample flow rate of 0.283 std L min$^{-1}$, which equals MDC = 0.236 std L$^{-1}$.

## 2.5   Quality control

The algorithm to derive the INP concentration also performs a quality control. When a deviation from the set conditions (see below) is observed, the data is stored normally but a flag is added to the measurement. The evaluation of the flag and a potential

exclusion of the data needs to be done by a researcher during post processing. Deviations are flagged (i) if the mean temperature of either wall is off by more than a predefined value (here $\pm 0.15$ K), (ii) one of the two MFCs reports a deviation between the set and the measured flow rate by more than $\pm 50$ std mL min$^{-1}$, (iii) the chiller reports an error, (iv) the pressure within the chamber is different by more than 50 hPa from the ambient pressure or (v) the water reservoir, used to rewet the chamber walls, is below a defined threshold (approx. 100 mL).

**3   Results**

## 3.1   Accuracy

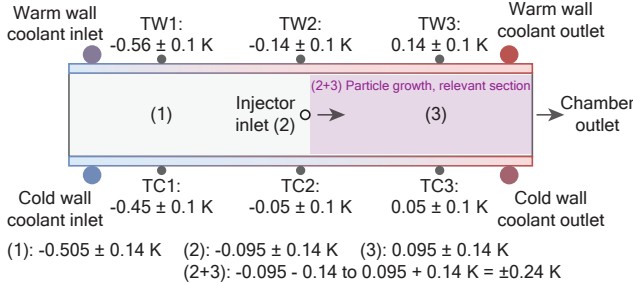

**Figure 7.** Schematic of uncertainty in temperature measurement showing a side view of HINC-Auto. TW and TC refers to one of six thermocouples installed on the warm wall and cold wall, respectively. Positions (1), (2), (3) and (2+3) indicate the temperature uncertainty of the center lamina. . For each of the positions (1), (2), and (3) we have the $\pm 0.1$ K form the warm and $\pm 0.1$ K from the cold wall thermocouple, thus $\pm\sqrt{0.1^2 + 0.1^2}$ K = $\pm 0.14$ K

Four main parameters characterize the INP concentration measured: temperature, supersaturation, particle count and volume flow. The thermocouples have an uncertainty of $\pm 0.1$ K and are calibrated measuring the melting of H$_2$O and Hg, in close agreement with the ITS-90 (the official protocol of the international temperature scale). The measured relative (compared to

set point $T$) temperature variation across the warm and cold wall is -0.56 / +0.14 K and -0.45 / +0.05 K at $T = 243$ K and $S_w$ = 1.04, respectively (see Figure 7). However, on each wall only the two thermocouples close to the injector (TW2/TC2) and the chamber outlet (TW3/TC3) are used to calculate the mean wall temperature. The relative variation therefore decreases to $\pm 0.14$ K (at the warm wall) and $\pm 0.05$ K (at the cold wall) for a center nominal temperature of $T = 243$ K at $S_w = 1.04$. In

either case, the temperature increases in the direction of the air flow, because of the parallel-flow setup of the cooling liquid (see Figure 2b). Subsequently, the uncertainty of the center lamina is -0.095 K for the relative variation plus $\pm0.14$ K for the thermocouples uncertainty at location (2) and 0.095 K $\pm$ 0.14 K at location (3). Therefore, the resulting total temperature variation in the section relevant for particle nucleation or activation and growth between the two cooling walls is $T$ $\pm0.24$ K. The accuracy of the supersaturation within HINC-Auto relies indirectly on the measured temperature, too, since the wall temperatures define the supersaturation. In addition, the vertical position of the aerosol layer determines the $T$ and $S$ experienced by the particles. Thus, the temperature uncertainty translates together with the maximal displacement of the particles within the center lamina with a sheath to sample flow ratio of 9:1 resulting in a supersaturation uncertainty of $S_w$ +0.007 and -0.009 and a total temperature uncertainty of $\pm1.11$ K at nominal $T$ = 243 K and $S_w$ = 1.04. This corresponds to a experienced range of 1.03 $\leq S_w \leq$ 1.05. According to the manufacturer, the used OPC can count 4995 particles per second, and simultaneously classify their optical size and place them in one of 6 user-defined size bins, with an overall accuracy of $\pm10\%$ to the calibrated aerosol. The sheath air MFC has a standard deviation of $\sigma$ = 1.07%, and the MFC downstream of the OPC a standard deviation of $\sigma$ = 0.25%. The CPC used for validation experiments has a counting uncertainty of $\pm10\%$ which yields in a relative uncertainty in the reported $AF$ of $\pm14\%$.

## 3.2 Validation

An overview of the validation of HINC-Auto is shown in Figure 8. Ammonium sulfate ($(NH_4)_2SO_4$), ammonium nitrate ($NH_4NO_3$) and sodium chloride (NaCl) were sampled from a aqueous solution with 0.1 mol L$^{-1}$, atomized, dried using a diffusion dryer to $S_w \leq 0.002$ at 20 °C and size selected to a mobility diameter $d_m$ = 200 nm by a differential mobility analyzer (DMA, TSI 3082, sheath flow set to 8 L min$^{-1}$, sample flow 1.3 L min$^{-1}$). A CPC (TSI 3787) measured the particle concentration in parallel to HINC-Auto to obtain $AF$. The particle concentration was targeted to 200 cm$^{-3}$. The injector position in HINC-Auto was kept constant yielding in between $\tau$ = 8 (273 K) and 10 seconds (218 K) residence time, depending on the temperature of the mid lamina. The experiments were conducted with constant mid lamina temperature while the super saturation is increased with $\Delta S_w$ = 0.02 min$^{-1}$. The onset of activation in the 1 μm-channel is reported for when the signal exceeds the background noise levels.

$NH_4NO_3$ is used to test cloud droplet formation at $T \geq$ 235 K and homogeneous freezing of solution droplets at $T \leq$ 235 K. Cloud droplet formation within the uncertainty range of HINC-Auto is detected at water saturation for 235 K $\leq T \leq$ 263 K. At the lower end of the temperature spectrum, the activation onset is more sudden (see Figure A5) and starts lower than water saturation. For homogeneous freezing at 233 K the activation is steep at $S_w$ = 0.97, 0.017 lower than the parameterized homogeneous freezing onset of solution droplets for $d_0$ = 200 nm and $J = 10^{10}$ cm$^{-3}$s$^{-1}$ (Koop et al., 2000b), but within the theoretically reported value considering uncertainty. At 228 and 222 K, the measured activation agree well with the parameterized homogeneous freezing onset. The deliquescence of ammonium sulfate ($(NH_4)_2SO_4$) was observed at $S_w$ = 0.88 (at 234.9K) to 0.86 (at 245.1K) within the range of uncertainty of literature values (Braban et al., 2001) as shown in

Figure 8. The deliquescence of sodium chloride (NaCl) lies within the range of uncertainty of measured values by Koop et al. (2000a).

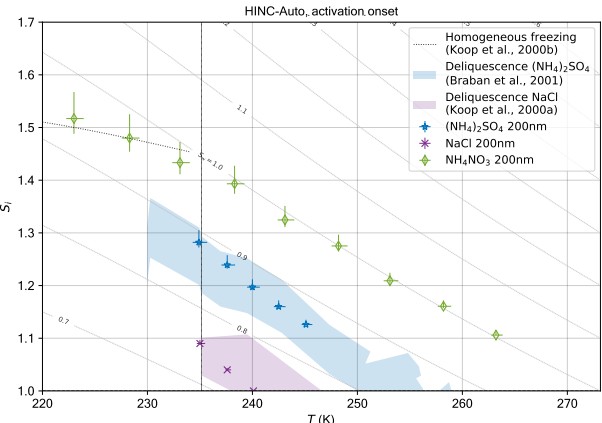

**Figure 8.** Data from experiments in HINC-Auto and comparison to values from literature for cloud droplet formation and homogeneous freezing onset with ammonium nitrate, and deliquescence of ammonium sulfate and sodium chloride. Reported is the activation onset when the signal increases above the background noise levels. Dashed line: homogeneous freezing onset of solution droplets for $d_0$ = 200 nm and $J = 10^{10}$ cm$^{-3}$s$^{-1}$ (Koop et al., 2000b).

### 3.2.1 Improvement in precision

Figure 9 shows the activation curves of ammonium nitrate size selected to a mobility diameter of $d_m$ = 200 nm and measured at $T$ = 233 K with HINC-Auto compared to measurements performed with HINC. The sample preparation is as described in section 3.2 for both chambers with a lower DMA sheath flow of 5 L min$^{-1}$ and a higher sample flow of 1.6 L min$^{-1}$ to feed both chambers and the CPC. This resulted in a broader transfer function within the DMA and consequently more larger and multiple charged particles penetrating the size selection. Due to the hygroscopicity of ammonium nitrate, the multiple charged particles are detected in the $\geq$ 1 $\mu$m OPC size bin after hygroscopic growth at $S_w$ < 0.98. In comparison, measurements in section 3.2 and Figure A5 use a narrower DMA transfer function and show a lower activated fraction below $S_w$ < 0.98 than in the experiment with the broader transfer function. The injector position was chosen to result in residence times of $\tau \approx 9$ seconds. The standard sheath to sample flow ratio of HINC-Auto was adjusted to 12:1 to be equal to the ratio used in HINC in order to compare the performance of the two chambers (note that the standard sheath to sample flow ratio of HINC-Auto is 9:1. See Section 2.2). HINC-Auto shows an improved precision compared to HINC. We attribute the improvement to the use of the mesh and, subsequently, the more uniform flow within HINC-Auto compared to HINC without the mesh. For measurements in the field with HINC, a defined supersaturation (e.g. $S_w$ = 1.04), temperature and OPC-size bin (e.g.$\geq$ 4 $\mu$m) is used to quantify INPs. Therefore, fluctuations in the activation precision of the $\geq$ 4 $\mu$m size bin can lead to uncertainties in INP concentrations. In the example of HINC, this is equivalent to more than one order of magnitude, thus an improved precision improves the

quality of the INP measurements. In addition, particle sedimentation, as expected by theory (see section below), is visible in the activation curves of HINC-Auto at $S_w \geq 1.02$ (see Section 3.2.2).

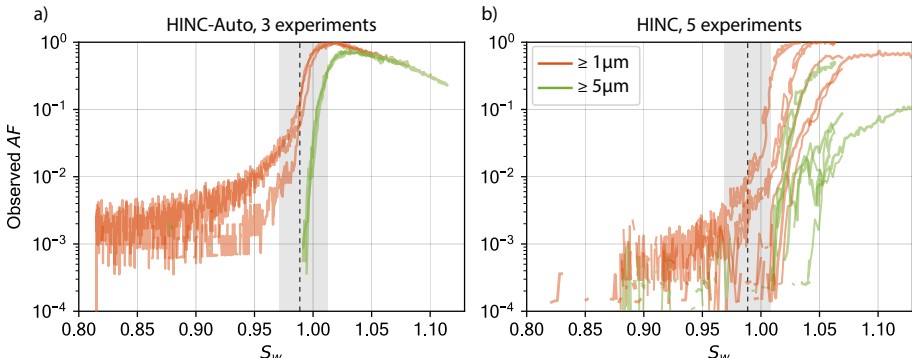

**Figure 9.** Activation curves of ammonium nitrate in HINC-Auto (a) and HINC (b) at $T = 233K$ with identical particle residence times of $\tau \approx 9$ sec.; the vertical dashed line indicates the expected homogeneous freezing onset of solution droplets for $d_0 = 200$ nm and $J = 10^{10}$ cm$^{-3}$s$^{-1}$ (Koop et al., 2000b). Sizes indicate what fraction of all particles entering the chamber grow to or beyond the indicated size. Grey shading refers to chamber uncertainty (see Lacher et al. (2017) for details on HINC and Section 3.1 for details on HINC-Auto).

### 3.2.2 Sedimentation study

The 2D diffusion model can also be used to calculate the diffusional growth of liquid and solid hydrometeors and their subsequent sedimentation characteristics. An example is given in Figure 10a and compared to an experiment of silver iodide (AgI, Figure 10b) at $T = 243$ K with a residence time of $\tau = 13.7$ sec at $p = 965$ hPa. AgI samples were prepared by aqueous solutions of 0.01 mol L$^{-1}$ of potassium iodide and silver nitrate, where the silver nitrate solution was slowly added to the potassium iodide under constant stirring. This procedure favors the formation of $\beta$-AgI over $\alpha$-AgI (Brauer, 1965). After resting for 60 min, the top 80% (consisting only of a clear solution) of the yellow suspension was decanted and the equivalent of removed volume was added in ultra pure water. A brief swivel lofted the settled precipitate. The decanting procedure was repeated twice. The suspension was atomized, dried using a diffusion dryer to $S_w \leq 0.008$ at 20 °C and size selected to $d_m = 100$ nm by a DMA. For the simulation, the fraction of INPs has been set to 15% for the simulation to agree best with the experiment. The fraction of INPs depends on the fraction of ice active $\beta$-AgI particles within all particles (Marcolli et al., 2016), which also contain ice inactive $\alpha$-AgI particles and cannot be deduced by the 2D diffusion model, and therefore, needs to be prescribed. The model uses the size distribution after the DMA measured by a Scanning Mobility Particle Sizer (SMPS) setup (DMA, TSI 3081, with CPC, TSI 3772) as input for the particle initial sizes and also places the particles (N = 1000 for each, INPs and CCN) at uniformly distributed vertical positions within the sample lamina. The model assumes instant activation of the particles as soon as ice or water saturation are reached. Also Köhler-theory is not implemented. Ice crystals are assumed to be spherical.

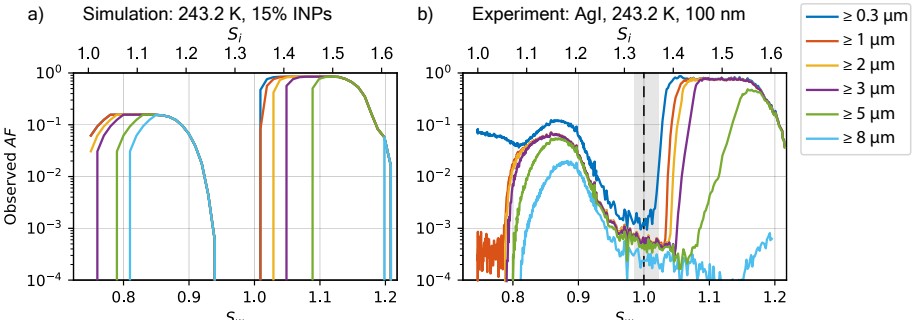

**Figure 10.** a) Simulated activated fraction curve as a function of $S_w$ with a prescribed fraction of INPs of 15% for $S_w < 1$ and b) measured activated fraction of $d_m = 100$ nm silver iodide (AgI) particles, both at $T = 243$ K, with a particle residence time of $\tau = 13.7$ sec at $p = 965$ hPa. Sizes stated in the legend indicate what fraction of all particles entering the chamber are activated and grow to or beyond the indicated size. Grey shading refers to chamber uncertainty around $S_w = 1.0$ (see Section 3.1 for details).

The model can capture the general trend of the experiment. The rapid onset of ice formation at $S_w \leq 0.8$ is not reproduced because this depends on the nucleation rate of the substance, which is neither parameterized nor simulated using molecular dynamics or similar approaches. However, the simulation demonstrates an upper bound of possible sizes, which is in agreement with the experiment. Where the ice growth is limited by the amount of supersaturation required for diffusional growth, e.g. at $S_w \geq 0.81$, the transition of ice crystals growing to a size $\geq 8$ µm is reproduced well. The sedimentation of the ice crystals is observed at $S_w \geq 0.87$ while the simulation predicts $S_w \geq 0.86$. The onset of liquid droplets is identical in the experiment for sizes $\leq 3$ µm, and delayed for larger sizes. Water vapor depletion is not likely the cause because running the experiment with a sample concentration of 20 or 1200 particles cm$^{-3}$ compared to the original 200 particles cm$^{-3}$ did not alter the slow growth of hydrometeors $\geq 5$ µm. We expect the delay to be present because of the missing implementation of Köhler-theory. The *AF* in the experiment as well as in the simulation is leveling off at $AF = 0.85$ for $1.02 \leq S_w \leq 1.13$. Ice crystals ($\approx 15$ % in the experiment) and supercooled droplets are continuously formed at these supersaturations, but ice crystals grow to such large sizes that they sediment and are not detected at the outlet anymore. Therefore, maximum droplet activation of all non-ice active particles is observed in this region. The sedimentation of droplets is observed delayed at $S_w \geq 1.16$ compared the model output at $S_w \geq 1.13$. This is likely a result of the delayed onset of liquid droplet formation seen in the experiment. While increasing the supersaturation the droplets remain too small, thus their size-dependent settling is delayed, too.

### 3.2.3 INP measurement

To assess HINC-Auto's capability of fully automated INP measurement, NX Illite was sampled in a controlled laboratory environment. The goal of this experiment is to assess how reliablly INPs can be detected over different INP concentrations and what their impact is on the ice layer endurance, thus deciding which rewetting intervals are needed and to test the overall automation of the chamber. HINC-Auto was, therefore, run autonomously except for a change in when to trigger the rewetting sequence. Using the software user interface, a command was sent to the chamber after the chamber was presumed to have

run dry. This has been inferred based on the decay of the reported INP concentrations. For the experiment, NX Illite was dry dispersed by a rotating brush generator (PALAS, RBG 1000) into a stainless steel aerosol chamber with V = 2.78 m$^3$, initially filled with dry, pure nitrogen. The volume within the aerosol chamber was actively stirred with a gold-plated fan (30 cm diameter). The observed particle number size distribution was 30 - 1000 nm (see Figure 11a). A common sample line from the aerosol chamber feeds a two way flow spitter, which connects HINC-Auto and a CPC (TSI 3778) using tubing with identical length and diameter. A burst of NX Illite was initially added to the aerosol chamber and continuously diluted by sampling 0.883 std L min$^{-1}$ (0.283 std L min$^{-1}$ for HINC-Auto and 0.6 std L min$^{-1}$ for the CPC) and adding 1.0 std L min$^{-1}$ of dry N$_2$. An additional bypass valve allowed to vent the excess N$_2$ before entering the aerosol chamber. HINC-Auto was run at mixed-phase cloud condition at $T$ = 243 K and $S_{\text{w}}$ = 1.04 to sample in the immersion freezing mode with a sheath to sample flow ratio of 9:1 and a residence time of $\tau \approx 8$ seconds. The chamber measured intervals of sample aerosols for 15 min and filtered background for 5 minutes. Figure 11b shows the measured INP and particle concentrations during the course of the experiment. Since the initial burst of NX Illite is suspended in dry, inert nitrogen, it is expected for the ice active fraction to remain fairly constant or decrease slightly over time. A slight decrease is expected when larger particles, which tend to be more ice active because of the higher surface area, sediment within the tank sooner than small particles. A SMPS setup (DMA, TSI 3082, with CPC, TSI 3787) measured the size distribution at the beginning of the experiment and after 62 hours (see Figure 11a). For the initial scan (solid line) the sheath flow was set to 5 L min$^{-1}$, and for the second scan to 2 L min$^{-1}$ (dashed line). At a lower sheath flow rate, the SMPS scan shifts to cover a larger range of particle sizes while limiting the scanning range at smaller particle sizes. Both measured size distributions were similar with a small shift towards larger particles for the later scan. This contradicts the assumption of large particles sedimenting more quickly than small particles, thus shifting size distribution towards smaller particle sizes with time. A reason could be due to particle coagulation in the stainless steel aerosol chamber, which causes the observed shift in size distribution towards larger particle sizes. The measured INP concentrations show a systematically repetitive trend: After each rewetting sequence the ratio between particle and INP concentration is approximately 1:1000. During the operation of the chamber the ratio remains fairly constant and decreases after some time. We expect the decrease to coincide with the depletion of the top wall ice layer. The higher the particle number concentration ($N$), the sooner the decrease is expected (due to consumption of water vapor mass), and observed. For atmospheric relevant conditions at the JFJ with 400 cm$^{-3} \leq N \leq$ 1000 cm$^{-3}$ the drop occurs after 8.5 hours, for 95 cm$^{-3} \leq N \leq$ 200 cm$^{-3}$ after 13 hours. $N$ is expected to contribute more in water vapor depletion than the INP number concentration because approximately 1000 times as many particles grow to liquid droplets of a size of $\approx$ 1.3 μm than INPs grow to $\approx$ 4.7 μm. At particle concentrations $\leq$ 30 cm$^{-3}$ and INP concentrations $\leq$ 30 std L$^{-1}$ the INP concentration starts to show noise which increases in relative magnitude with decreasing particle concentrations. A running mean of 4 subsequent measurements decreases the noise and allows the particle to INP ratio to remain until $\approx$ 6 INP std L$^{-1}$. Based on the above experiment we choose a rewetting time of 8 hours for field applications in remote areas such as Jungfraujoch.

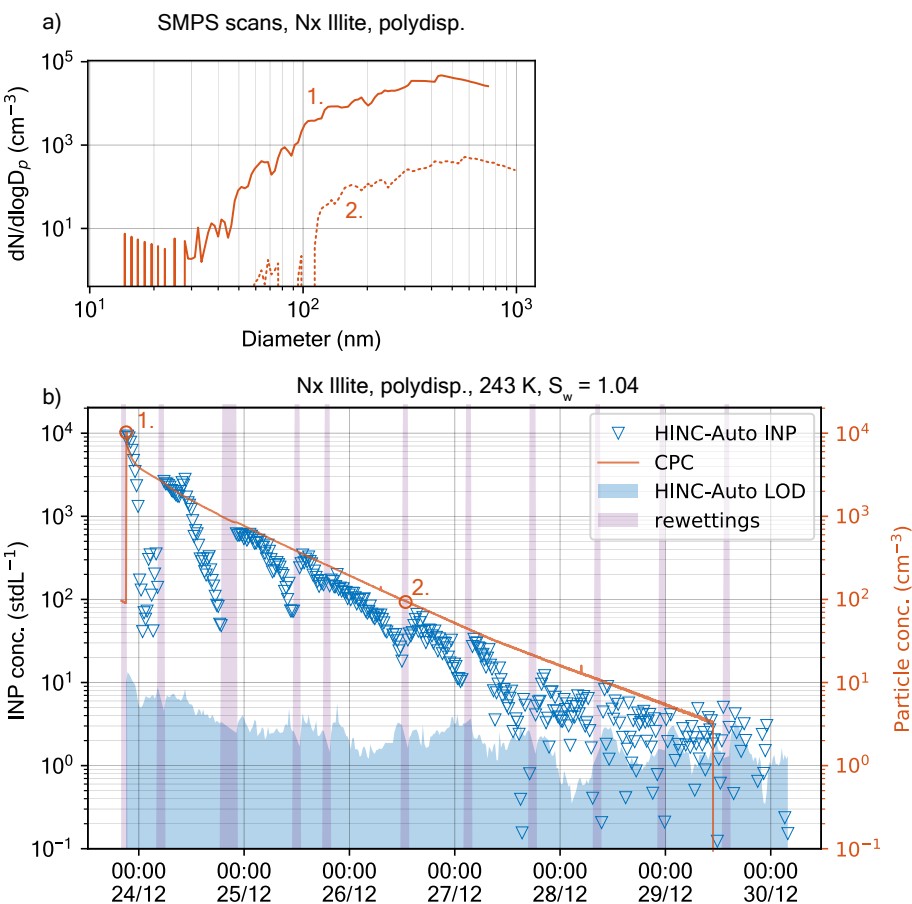

**Figure 11.** INP concentration in HINC-Auto as a function of time and NX Illite particle concentration. a) SMPS retrieved particle size distribution at time 1 (solid line) and 2 (dashed line). The DMA sheath flow was set to 5 (solid line) and 2 L min$^{-1}$ (dashed line). b) Measured INP and particle concentrations within the aerosol tank.

## 3.3 Application

The automation of HINC-Auto was tested during a field campaign in August 2019 on the JFJ. In this time, HINC-Auto was operational for 169 out of the 177 available hours (95.5%) and measured 463 measurements-background sequences of a duration of 20 minutes each (see Figure 12a). The maximum measurement coverage with HINC in a time window of 177 hours during previous field campaigns was 191 measurement sequences (see Lacher et al., 2018), also 20 minutes in duration (103 hours, 58%, see Figure 12b). The automation shows a clear improvement in the continuity of the measurements. However,

during the field campaign the LOD of HINC-Auto was appreciably higher than the LOD of HINC during previous field campaigns (median LOD HINC-Auto: 3.1 std L$^{-1}$, HINC: 1.25 std L$^{-1}$). Design changes implemented in a second field campaign started in February 2020 resulted in a median LOD of 1.37 std L$^{-1}$. It was observed that ice crystals and frost deposited within the cavity of the chamber outlet. We assumed for supercooled liquid droplets, which make up the majority

of the hydrometeor population exiting the chamber, to impact on the surfaces where the Swagelok fitting (to connect the OPC) is inserted into the PVDF frame. The change in inner diameter from the cavity within the PVDF frame ($d_i$ = 10.2 mm) to the fitting ($d_i$ = 3.3 mm) is like a step. The design changes included using a conical drill bit (20°) to smoothen the connection between the chamber outlet and the fitting. In addition, the Swagelok fitting is warmed with a 10 W heat pad during the rewetting procedure to support the evaporation of residual condensate or molten ice that does not drain due to gravity while the chamber is tilted. In the first month the chamber was operational for 698 out of 720 available hours (97.0%). A broken membrane of the recirculating pump caused 10 hours of downtime, the measurement of ramps at the beginning of the campaign resulted in 7 hours when the chamber was not performing INP measurement sequences. The remaining 5 hours were due to multiple software glitches which were fixed. In March 2020, HINC-Auto was operating more reliably with a total downtime of 3 hours because of software bugs. Bug fixes were implemented and let to an operation without any downtime in April 2020. Continuous INP measurements of HINC-Auto at the JFJ are planned to proceed. Live data can be accessed at https://www.psi.ch/en/lac/projects/last-72h-of-aerosol-data-from-jungfraujoch.

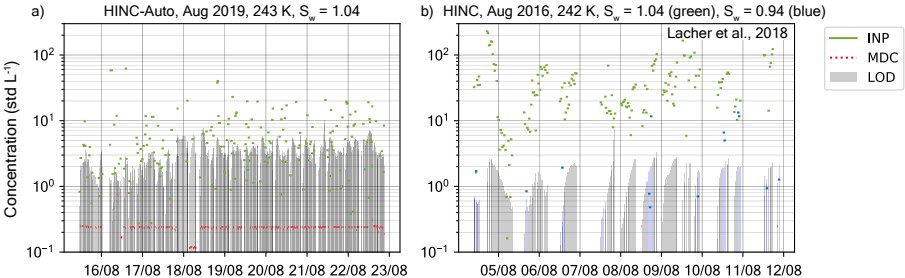

**Figure 12.** INP measurements at the JFJ over a period of 7 days with a) HINC-Auto at $T$ = 243 K and $S_\mathrm{w}$ = 1.04 and b) HINC at $T$ = 242 K and $S_\mathrm{w}$ = 1.04 (green markers) or $S_\mathrm{w}$ = 0.94 (blue markers) (see Lacher et al., 2018). Green/blue markers indicate the ambient INP concentration, the grey bar indicates the limit of detection (LOD) with a 62.3% (1 $\sigma$) confidence interval, red markers indicate measured INP concentrations below the minimum detectable concentration (MDC, 1 count during the sampling time). Every grey bar with a corresponding green/blue/red dot represents one measurement.

## 4    Conclusions

In this study the automated Horizontal Ice Nucleation Chamber (HINC-Auto) is presented, a continuous flow diffusion chamber to measure atmospheric ice nucleating particle number concentration. During a first field campaign, the chamber was operational fully autonomously for 95% of the time. Over a 90 day period of a second field campaign, HINC-Auto measured INP autonomously for 97% of the time. This time includes INP sampling time as well as periodically reoccurring rewetting procedures. To realize full autonomous operation, the rewetting procedure in HINC was automated. In HINC-Auto, a peristaltic pump remoistens the filter paper that coats the top (warm) wall within the chamber while a linear motor tilts the chamber thereby allowing excess water to drain. The bottom wall passively wets through diffuson of vapour from the cold wall. To maximize the sampling time, the duration of the rewetting procedure was reduced by reducing the warming and cooling time of the

445 chamber by using a lower thermal mass for the chamber walls. This was done by using aluminium compared to copper and by shortening the chamber. To maintain a well developed relative humidity profile within the sampling section in the shortened chamber, the sheath air is precooled using heat sinks mounted on the cold, bottom wall. Besides the automation, improvements to the flow uniformity were achieved by the integration of a mesh. The resulting activation curves during experiments with relative humidity ramps are steeper in expectation to theory and show a better reproducibility. In addition, the experiments

match the predictions of a 2D diffusion model to a high degree. Experiments with HINC-Auto for cloud droplet formation and homogeneous freezing onset with ammonium nitrate, and deliquescence of ammonium sulfate and sodium chloride showed good agreement with values found in literature. HINC-Auto is currently deployed for long-term continuous measurements of INP at JFJ and analysis after a year of data collection will be presented in a follow up study.

*Code availability.* The code used for simulations presented here is available upon contacting the authors.

*Data availability.* The data presented in this publication will be made available at DOI: 10.3929/ethz-b-000429220. Note by authors: Data will be made available upon publication

## Appendix A

### A1   CFD simulations

The CFD simulations were performed using Siemens PLM STAR-CCM+ © V13.04.010. A grid convergence study was con-

460 ducted prior to the simulations. A 96% convergence of the flow field was observed for the following settings, which were used for all subsequent studies. A tetrahedral mesh with a total count of $9.5 \times 10^6$ cells was generated with the surface remesher option and 5 prism layers with a stretching of 1.3. The base mesh size was set to 2 mm with a refinement of 0.25 mm around the injector and the heat sinks and a further refinement of 0.1 mm in the vicinity of the injector's slit. The physics model consisted of 3D turbulent Reynolds-Averaged Navier-Stokes equations, k-$\omega$ turbulence model, Lagrangian multiphase with water vapor

and air, steady-state, segregated fluid enthalpy, ideal gas, and gravity. Convergence was observed after $\sim 2000$ iterations with energy-residuals $< 10^{-5}$ as criterion. An unsteady simulation showed no instationary flow behavior and was consistent with the steady-state solution.

## A2 Particle Image Velocimetry

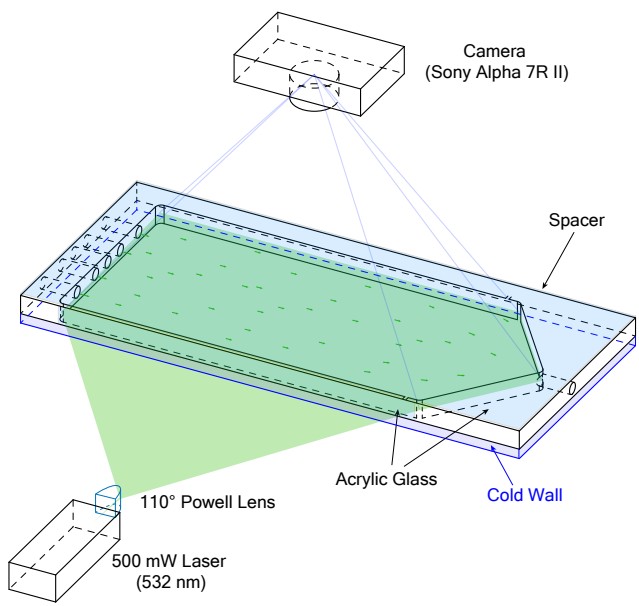

**Figure A1.** Schematic of the particle image velocimetry experiment with HINC.

Particle image velocimetry (PIV) has been used to validate the CFD-simulations. Figure A1 illustrates the experimental set-up using HINC. Dry NX Illite size-selected particles to a mobility diameter $d_m = 400\ nm$ have been used as tracer. Two experiments were conducted: first, the chamber without the injector. The particles were added to the sheath air. The goal was to sample the overall flow within HINC and deduce the 2D-velocity field. The injector was installed for the second experiment. Here, the areas of interest were to study the impact of the injector on the base-flow within the chamber, as well as the trajectories of non-sedimentating particles as they exit the injector. All results are for the purpose of validating the CFD-simulations. The image sequences were recorded by a SONY Alpha 7R II with a resolution of $\Delta_{low}^2 = 3840 \times 2160$ pixel and $\Delta_{high}^2 = 1920 \times 1080$ pixel. The camera captured images at $f_{low} = 25$ fps and $f_{high} = 100$ fps. A 500 mW-laser-diode with a wavelength of $\lambda = 532\ nm$ was used as continuous light source. The laser beam is deflected with a K9 110 degree Powell lenses into a straight line, and oriented such, that it illuminates a horizontal plane. The laser's vertical position was controlled by a stepper-motor and could be adjusted with a resolution of 0.05 mm. This allowed to scan the entire chamber and produce a 3D matrix containing the xy-planar velocity components. The laser-facing wall of the CFDC's spacer has been replaced by a planar transparent acrylic glass to allow the laser beam to pass. Also, the top cooling wall has been replaced by clear acrylic glass to allow the camera to capture the illuminated particles. The lower wall was chilled to 279.6 K, while the top acrylic's temperature was at ambient (297 K). The resulting temperature difference of 17.4 K is identical to when the chamber is operated at $T = 243$ K and $S_w = 1.04$. It is assumed that the absolute temperature has a minor impact on the flow, while it is important to mimic the relative temperature differences since it represents the difference in fluid densities. The post-

processing has been done using PIVlab (Thielicke and Stamhuis, 2014). Fast Fourier Transform linear window deformation technique was used as PIV algorithm with an interrogation area of 128 pixel and a step size of 64 pixel. A second pass an interrogation area of 64 pixel and a step size of 32 pixel followed. Figures A2 and A3 compare the flow field retrieved from the PIV experiments with the CFD-simulations. For the PIV experiments $low$-settings were used to capture the flow of the entire fluid domain, whereas the $high$-settings were used to extract flow features where the $low$-settings failed to produce a valid output. Nevertheless, in the areas after the sheath flow outlets the flow velocity was too high to be captures with the $high$-settings. Also, close to the spacer walls, no velocity information could be computed, because of the Fast Fourier Transform window deformation technique. Quantitatively the CFD simulations are able to capture the flow structures measured during the experiment. Quantitatively, the correlation of the $u, w$-velocity components all voxels resolved by the experiment with the $u, w$-velocity components of the simulation with identical locations were $r_{HINC} = 0.94$ and $r_{mesh} = 0.95$ for the experiment without and with mesh, respectively. Least correlation was observed in the areas of the sheath flow jets where high flow velocities are present. Also, the simulated particle trajectories were qualitatively stringent with the experiment. Therefore, the CFD simulations were deemed valid.

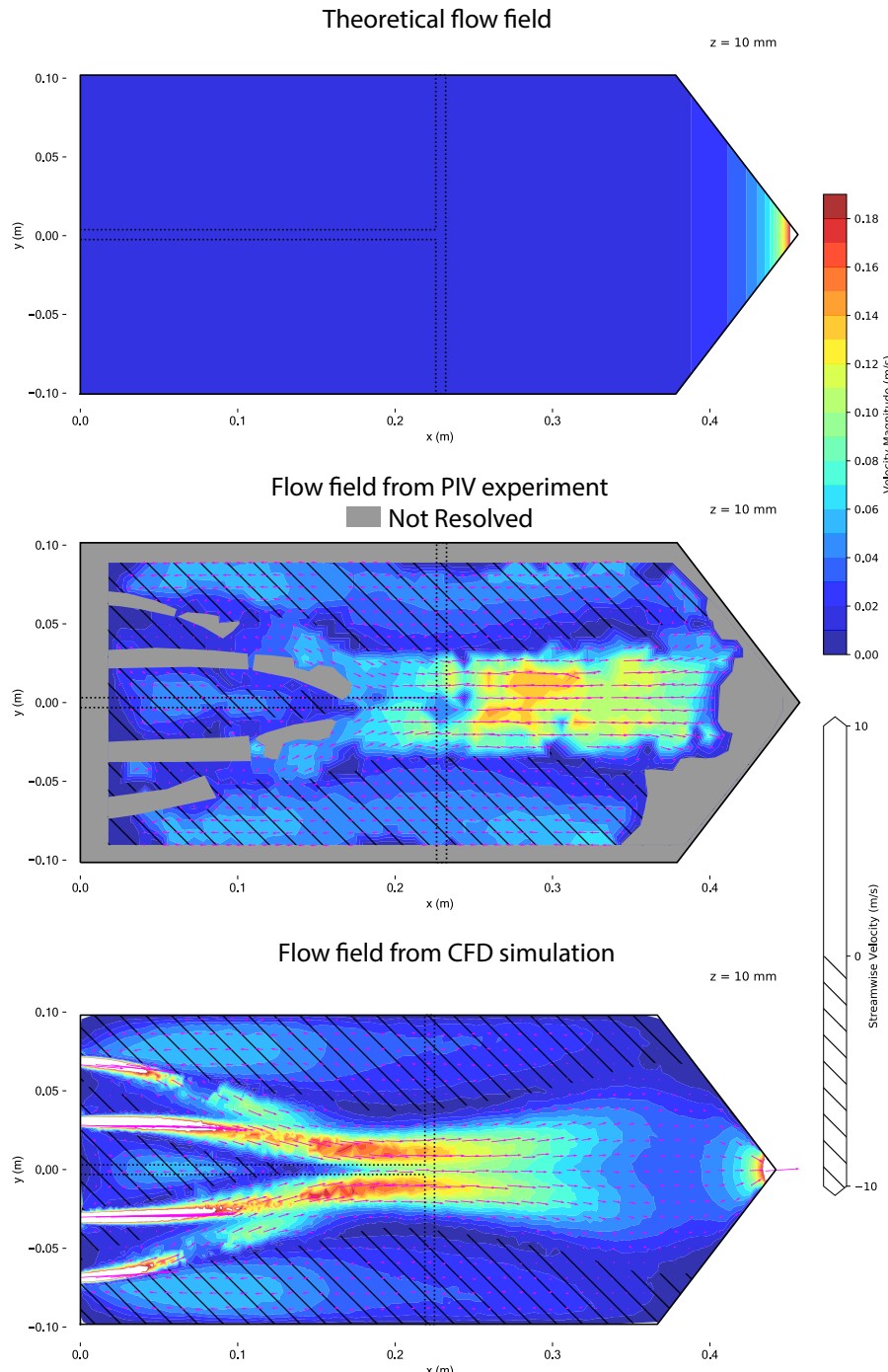

**Figure A2.** Contour plot of flow velocities magnitudes at a horizontal plane of the HINC cavity at mid-height. The chamber is set to $T = 243$ K and $S_w = 1.04$. Theoretical flow field (top), flow field measured using PIV (center) and simulated flow field from CFD (bottom). Arrows indicate flow direction, the hatching marks regions with reversed flow. The dashed line represents a possible position of the injector.

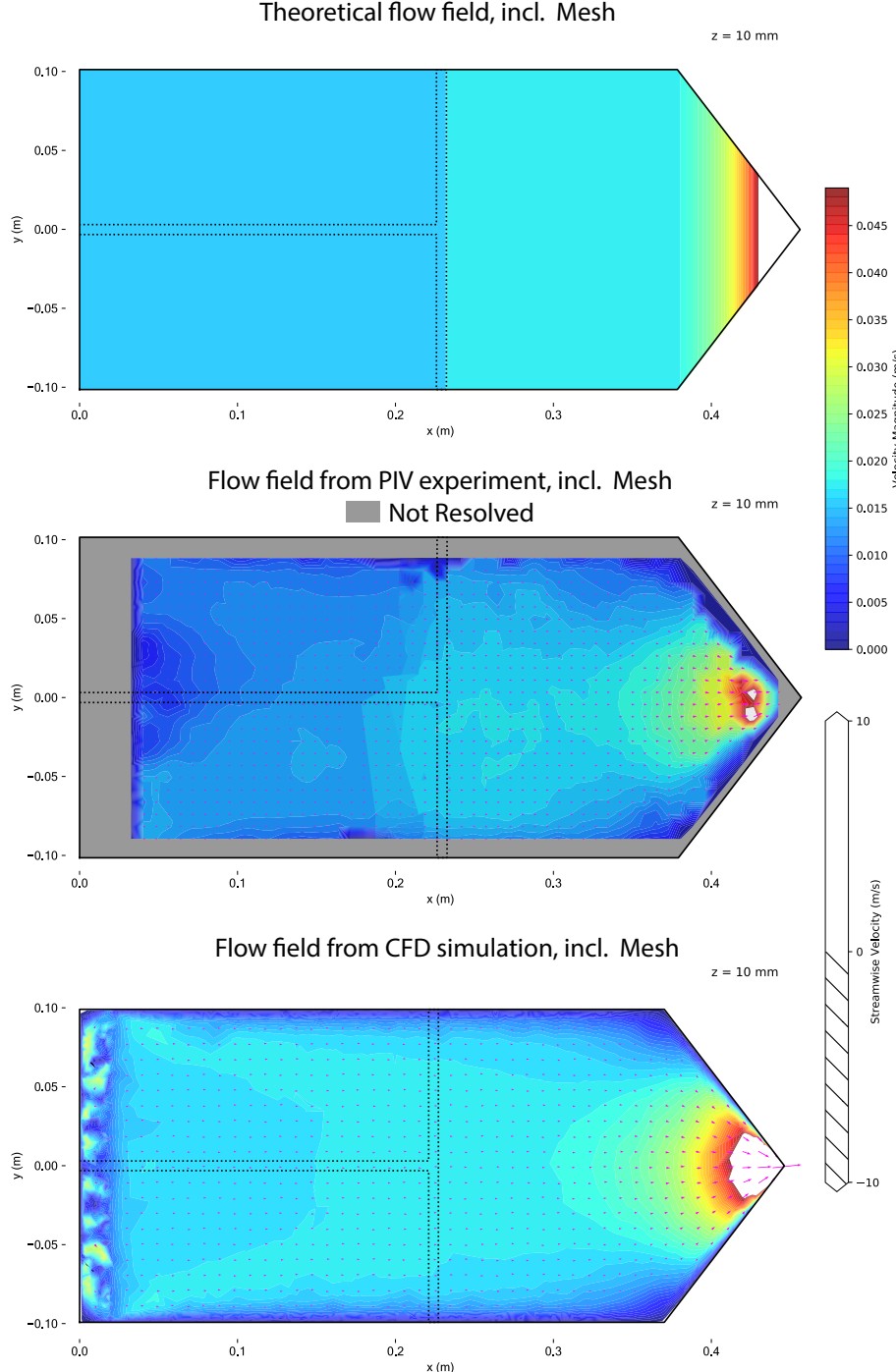

**Figure A3.** Contour plot of flow velocities magnitudes at a horizontal plane of the HINC cavity at mid-height including a mesh installed at $x_{inj} = 0.022 \; m$. The chamber is set to $T = 243$ K and $S_w = 1.04$. Theoretical flow field (top), flow field measured using PIV (center) and simulated flow field from CFD (bottom). Arrows indicate flow direction, the hatching marks regions with reversed flow. The dashed line represents a possible position of the injector.

## A3  2D diffusion model

A numerical diffusion model has been developed to simulate the water vapor saturation field within HINC-Auto. The output of the model is a 2D vertical plane of a horizontal CFDC and its prevailing temperature and water vapor saturation field. Assuming a two-dimensional flow along the chambers length, for the prevalent conditions with Reynolds number $Re = 6.82 \times 10^2$ between two long plates the analytical solution of the Navier–Stokes equations describes a Poiseuille flow according to equations (A1) and (A2):

$$u_{\infty_{bulk}}(x) = \frac{\dot{V}}{A_{yz}(x)} \tag{A1}$$

$$u(x,z) = \frac{3}{2} u_{\infty_{bulk}}(x) \left( 1 - \frac{\left(z - \frac{h}{2}\right)^2}{\left(\frac{h}{2}\right)^2} \right) \tag{A2}$$

Where $x, y, z$ are the coordinates along the length, width and height of the chamber. The origin is located on the bottom cooling wall where the sheath-air enters at mid-width. $u_{\infty_{bulk}}(x)$ is the bulk velocity along the chamber's length, $V$ is the preset volume flow of air, $A_{yz}(x)$ is the vertical cross-section of the chamber's cavity which varies along the length of the chamber, $u$ is the stream wise velocity component and $h$ is the total height of the chamber. According to equation (A2) the center lamina has a 50% higher flow velocity than the bulk flow velocity. The particle's theoretical residence time $\tau_{theory}$ is calculated using equation (A3):

$$\tau_{theory}(x_{inj}) = \int\limits_{x_{inj}}^{x_{outlet}} \frac{x}{u_{\infty_{bulk}}(x)} dx \tag{A3}$$

Where $x_{inj}$ is the injector's $x$-position and the outlet's $x$-position $x_{outlet}$.

In CFDCs the temperature and water vapor concentration of the air flow equilibrates from their initial states to form a water vapor supersaturated region. To calculate the sedimentation of particles as well as estimate the particle's final size the two underlying diffusion processes have to be described. Firstly, the heat flux between the warm and cold wall. We assume in accordance to Saxena et al. (1970) no radiation. Although the Nusselt number $Nu = 15.3$ propose otherwise, the calculations are simplified by considering thermal conduction only and neglecting the forced convection. The heat conduction within the ice layer can be neglected (Biot number $Bi = 7.27 \times 10^{-5}$). The calculated numbers are computed for HINC-Auto, the corresponding values of the chamber of Rogers (1988) are $Nu = 22.3$ and $Bi = 1.14 \times 10^{-4}$ for $T_{mid} = 253.15\ K$ and $\dot{V} = 4$ std L min$^{-1}$. Besides, the latent heat released or absorbed by the particles or by the wall's ice layer is neglected for calculation of the chamber's internal temperature field. The former is taken into consideration during the diffusion growth calculations (Rogers and Yau, 1989). This leads to the diffusion of heat in air by changing their kinetic energy, according to the heat equation (A4) (Amelin, 1967):

$$\frac{\partial T_{air}}{\partial t} = \frac{\lambda_{air}}{c_{p_{air}} \rho_{air}} \frac{\partial^2 T}{\partial x^2} \tag{A4}$$

Where $T_{air}$ is the temperature of the air, $t$ is the temporal variable, $\lambda_{air}$ is the thermal conductivity of air, $c_{p_{air}}$ is the heat capacity of air at a constant pressure, $\rho_{air}$ is the density of the air and $x$ is the spatial coordinate.

The second diffusion process is the diffusion of water vapor according to Fick's law (Fick, 1855) in equation (A5):

$$\frac{\partial \mu_{H_2O}}{\partial t} = D_{H_2O/air} \frac{\partial^2 \mu_{H_2O}}{\partial x^2} \tag{A5}$$

Where $C_{H_2O}$ is the concentration of water vapor, $D_{H_2O/air}$ is the diffusivity of water vapor in air and $\mu_{H_2O}$ is the chemical potential of the water vapor.

Saxena et al. (1970) states in equation (A5) to use the water vapor concentration $C_{H_2O}$ instead of the chemical potential, yet, it was chosen to use the water vapor partial pressure $e$ in accordance with Rogers (1988). Fundamental theory often suggests the diffusion to be driven by the difference in chemical potential (Sutherland, 1905; Einstein, 1905). Diffusion based on partial pressure or concentration results in different supersaturations than diffusion based on chemical potential. This has has a profound impact on the operating conditions in CFDCs, since the partial pressure is a function of the concentration as well as of the temperature, as seen in the rearranged ideal gas law, (A6), and changes the supersaturation in a non-linear fashion:

$$e = \frac{C_{H_2O} RT}{M_{H_2O}} \tag{A6}$$

To calculate $S$ the equilibrium saturation vapor pressure of water and ice according to Murphy and Koop (2005) is used. For the 2D diffusion model, both heat, and diffusion equations are solved numerically using the forward Euler method. The velocity is prescribed using equation (A2). The 2D fluid domain is modeled using a mesh with a grid spacing of 1 mm (7500 nodes in total). The steady state simulation converged after $\leq 100$ iterations. Buoyancy has been neglected because it showed to have minor effect on the supersaturation field within horizontal CFDC. The model output has been compared versus a validated CFD simulation as visualized in Figure A4.

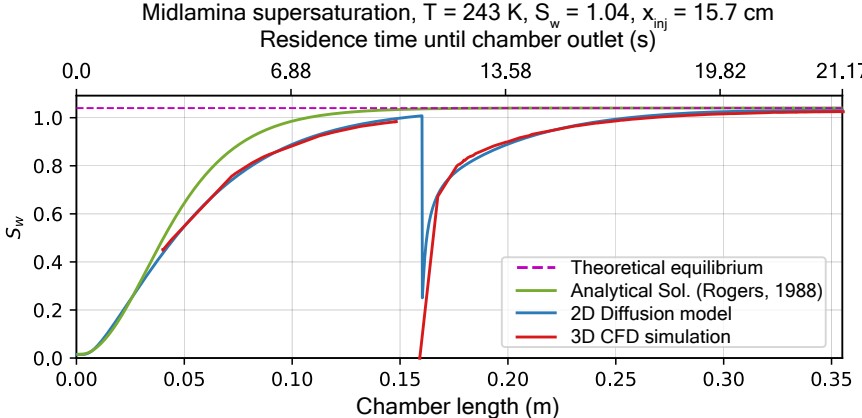

**Figure A4.** Comparison of the midlamina water supersaturation at $T = 243$ K, as calculated using the analytical formula (Rogers, 1988), simulated using the 2D diffusion model and the 3D-CFD simulation. Injector position $x_{inj} = 15.7$ cm.

Diffusional growth is calculated according to Rogers and Yau (1989) with the latent heat of sublimation of ice, and latent heat of vaporization of supercooled water according to Murphy and Koop (2005). $T$ and $S$ are variable and feed in from the 2D diffusion model corresponding to the particle's current horizontal and vertical position within the chamber. The diffusional growth of the hydrometeors assumes activation when saturation with respect to ice or water is exceeded.

## A4 Ammonium nitrate experiments

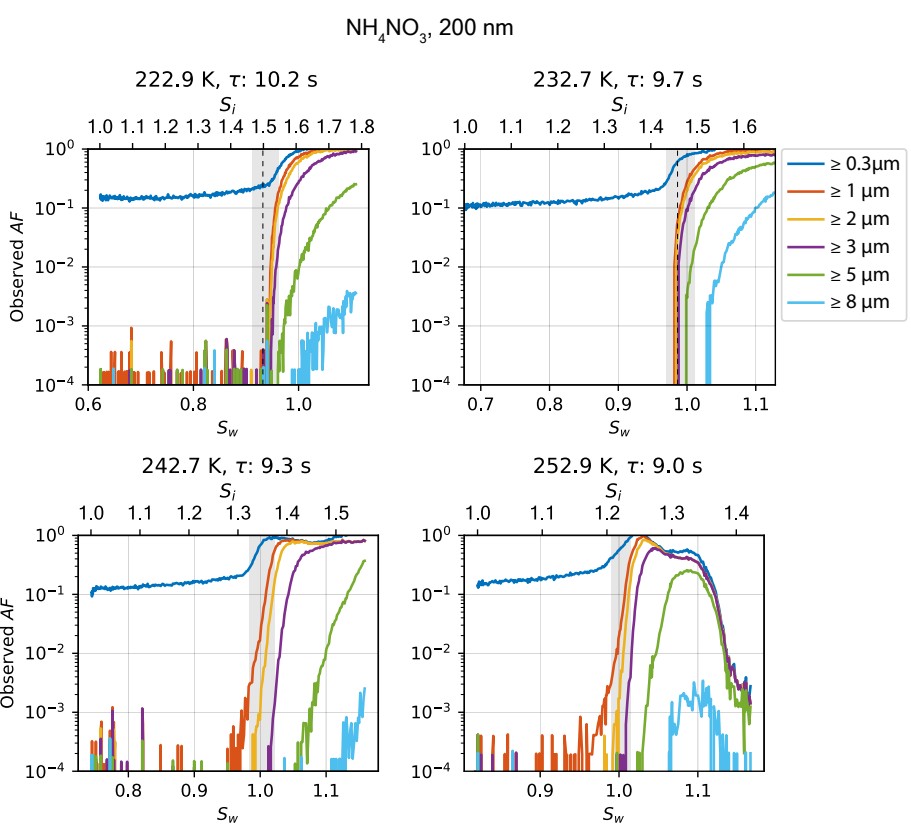

**Figure A5.** Relative humidity ramps showing the increase in *AF* during deliquescence and cloud droplet formation of 200 nm ammonium nitrate particles at various temperatures. Grey shading refers to chamber uncertainty (see Section 3.1 for details).

## A5 Ammonium sulfate experiments

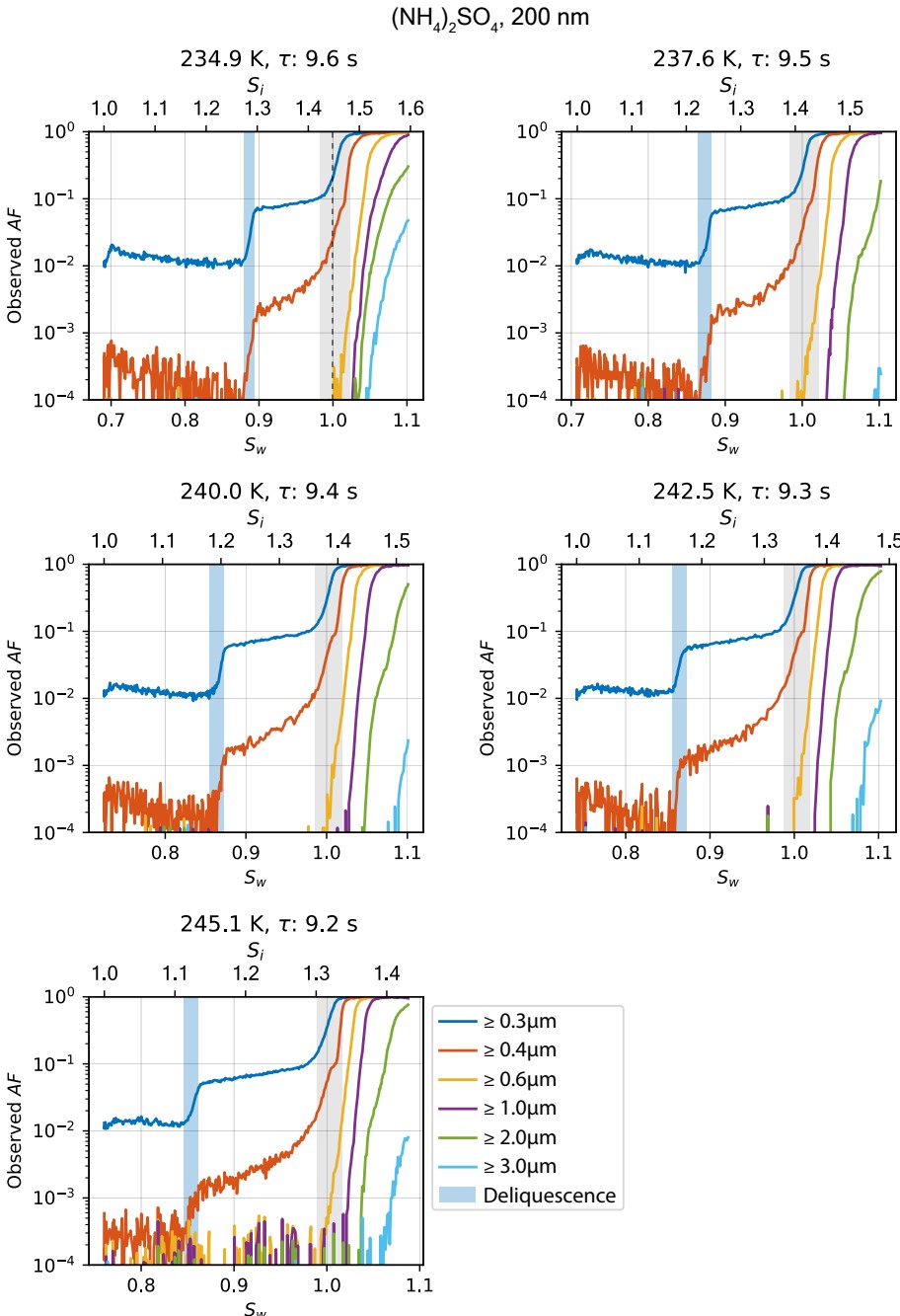

**Figure A6.** Relative humidity ramps showing the increase in *AF* during deliquescence and cloud droplet formation of 200 nm ammonium sulfate particles at various temperatures. Light blue shading highlights range of observed deliquescence, grey shading refers to chamber uncertainty (see Section 3.1 for details).

## A6    Sodium chloride experiments

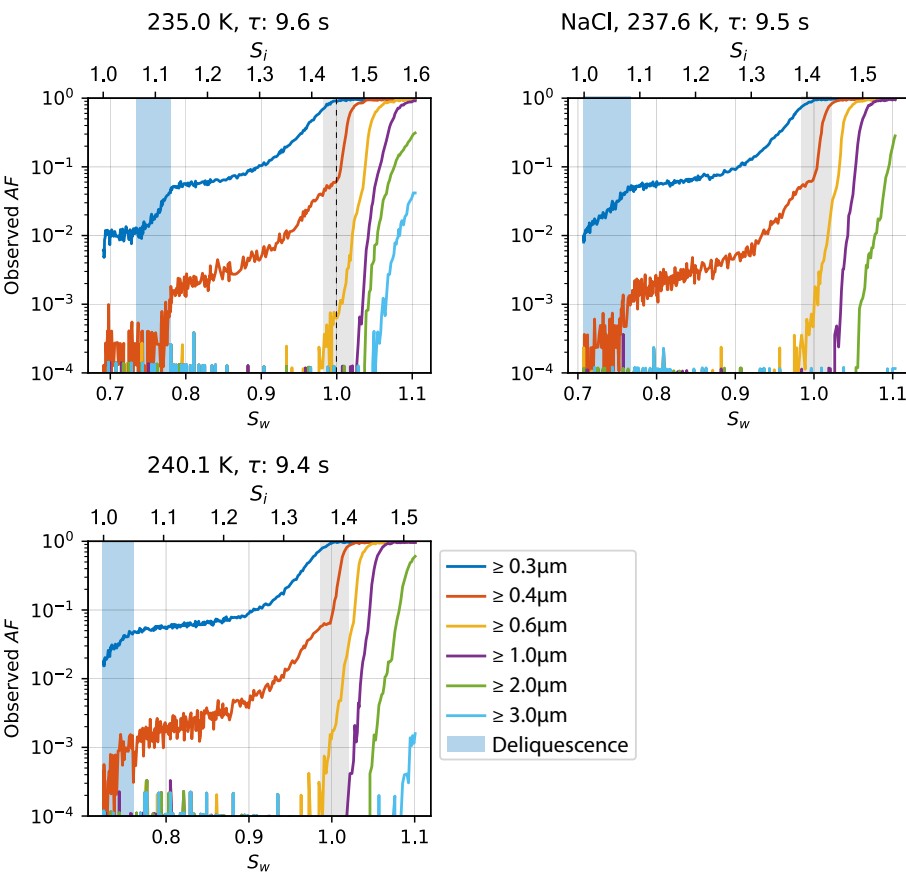

**Figure A7.** Relative humidity ramps showing the increase in *AF* during deliquescence and cloud droplet formation of 200 nm sodium chloride particles at various temperatures. Light blue shading highlights range of observed deliquescence, grey shading refers to chamber uncertainty (see Section 3.1 for details).

## A7    Guided user interface

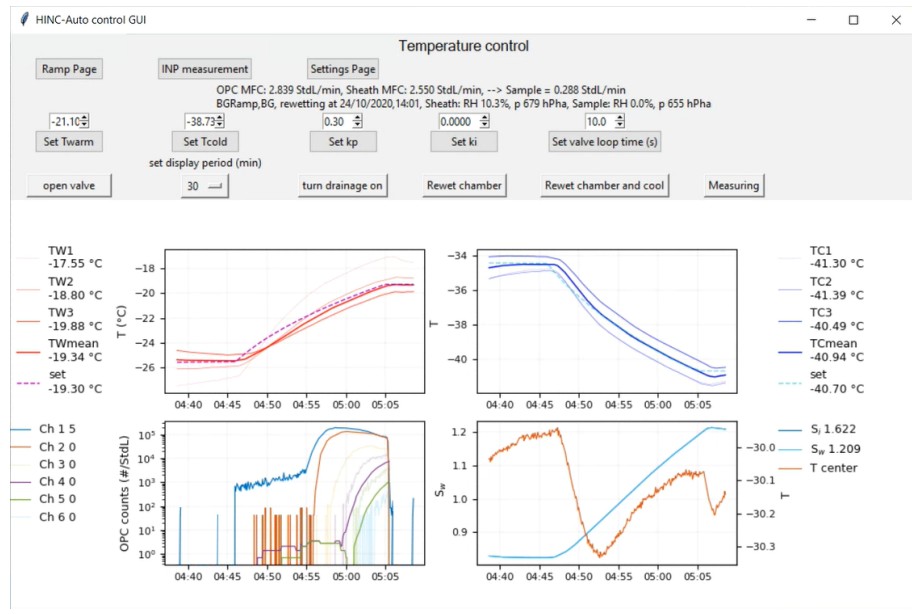

**Figure A8.** Main window of the guided user interface used to control HINC-Auto. Top row buttons: **Ramp Page** leads to the window to run $S_w$- or $T$-ramps (see Figure A9), **INP measurement** leads to the window to run INP measurements (see Figure A10) and **Settings Page** leads to the window to change additional settings (see Figure A11). **OPC MFC** shows the current flow rate of the OPC mass flow controller (MFC), as reported by the MFC, **Sheath MFC** analogously shows the flow rate of the sheath flow MFC, **Sample** reports the difference between both MFC, which corresponds to the current sample flow rate. **BGRamp** shows the target state of the chamber, in this case "measuring the background after or before a $RH$- or $T$-ramp, **BG** shows the state in which the chamber currently is, **rewetting at** indicates the time of the next planned rewetting procedure (UTC), **Sheath** indicates the $RH$ and pressure $p$ of the sheath air as it enters the chamber (measured at mid-height within the chamber upstream of the mesh, thus at the same temperature as the chamber is set to) and **Sample** indicates the $RH$ and pressure $p$ of the sample air (just downstream of the sample diffusion dryer. Center row buttons: **Set Twarm** sets the target temperature of the warm wall, **Set Tcold** sets the target temperature of the cold wall, **Set kp** and **Set ki** set the proportional gain ($k_p$) and integral gain ($k_i$) of the PI-controller to control the warm wall temperature, **Set valve loop time (s)** defines the duration of a loop to actuate the warm wall solenoid valve. Bottom row buttons: **open valve** manually opens and closes the warm wall solenoid valve, **Set display period (min)** defines the amount of historic data to be shown in the four graphs below, **turn drainage on** manually actuates the pump to drain the chamber during rewetting, **Rewet chamber** executes the rewetting procedure, **Rewet chamber and cool** executes the rewetting procedure and cools the chamber back down to the set temperature of the chamber before the button was activated, **Measuring** changes the chamber's state and valves to sampling or background measurement. The top left graph shows the warm wall temperature (all three thermocouples, the computed mean wall temperature and the set temperature). The top right graph shows the cold wall temperature (all three thermocouples, the computed mean wall temperature and the set temperature). The bottom left graph shows the counts reported by the OPC for each of the six size-bins/ channels (here CH 1 set to $\geq 0.3\ \mu m$, CH 2 set to $\geq 1\ \mu m$, CH 3 set to $\geq 3\ \mu m$, CH 4 set to $\geq 4\ \mu m$, CH 5 set to $\geq 5\ \mu m$, CH 6 set to $\geq 6\ \mu m$). The bottom right graph shows $S_w$ and $T$ for the center lamina (calculated).

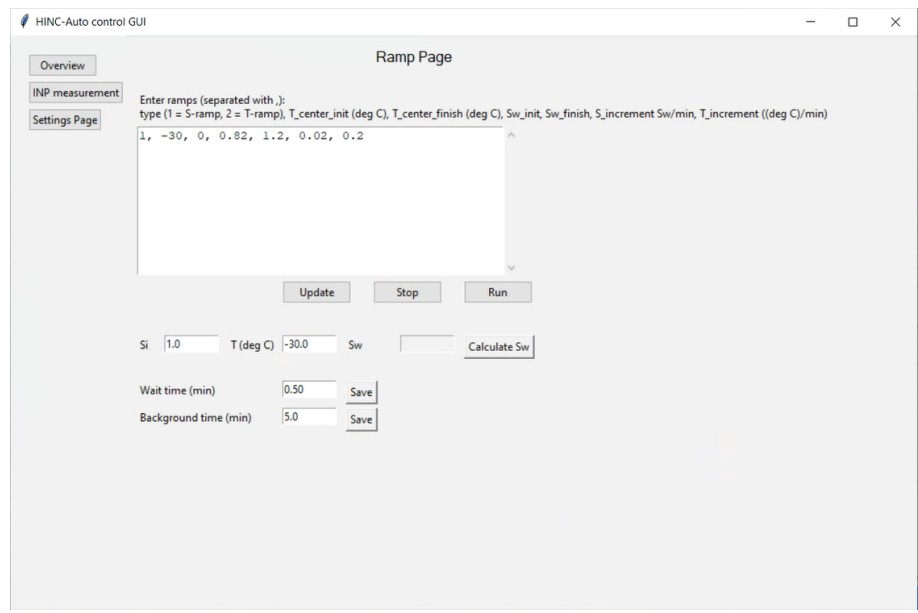

**Figure A9.** Window of the guided user interface used to control *RH*- or *T*-ramps with HINC-Auto.

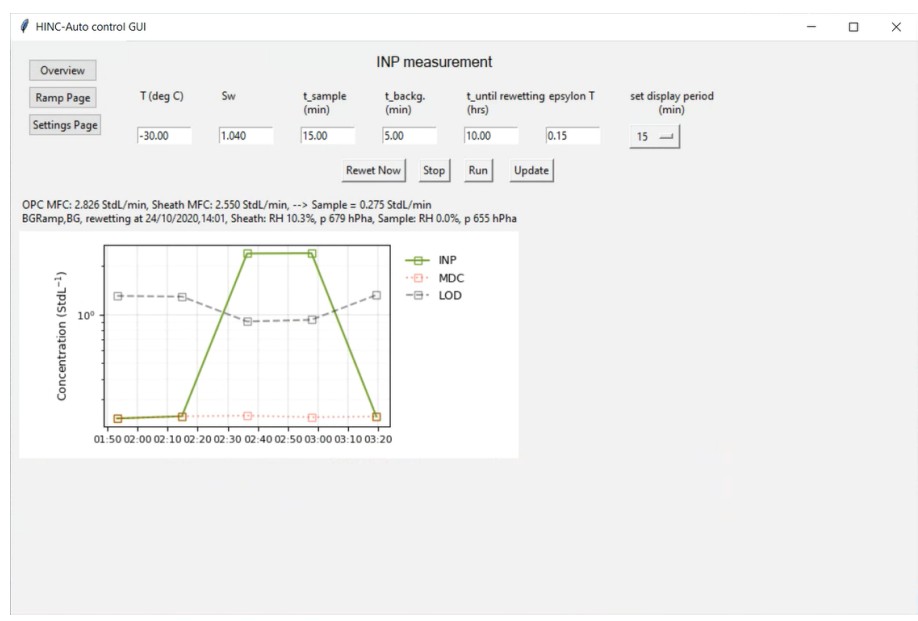

**Figure A10.** Window of the guided user interface used to control INP measurements with HINC-Auto.

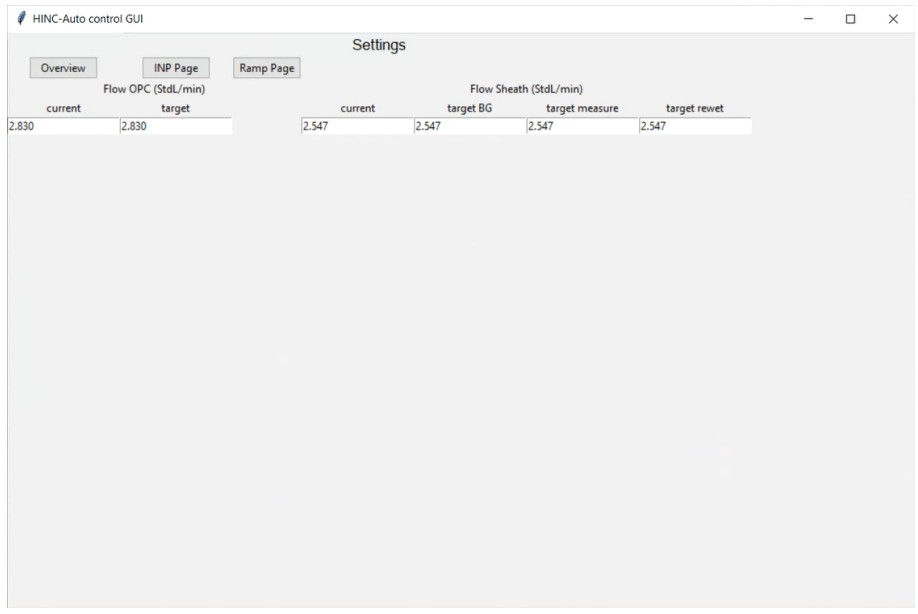

**Figure A11.** Window of the guided user interface used to adjust additional settings with HINC-Auto.

*Author contributions.* CB conducted experiments with input from ZAK. CB designed the HINC-Auto chamber with input from ZAK. CB
designed and conducted PIV and CFD experiments. CB developed the 2D diffusion model. CB analysed the data and prepared the figures
with input from ZAK. CB and ZAK interpreted the data. CB wrote the manuscript with input from ZAK. ZAK conceived the idea, supervised
the project and obtained funding.

*Competing interests.* The authors declare that they have no conflict of interest.

*Acknowledgements.* This research was funded by the Global Atmospheric Watch, Switzerland (MeteoSwiss GAW-CH+ 2018–2021). We
thank Prof. Dr. Ulrike Lohmann for her support and enthusiasm. We acknowledge Jörg Wieder, Zane Dedekind, Dr. Fabian Mahrt, and Dr.
Carolin Rösch for useful discussions. For technical support and fabrication we would like to thank Dr. Michael Rösch and Marco Vecellio,
whose expertise greatly helped to improve the instrument. We thank the editor Mingjin Tang and two anonymous referees for their valuable
contribution to the manuscript.

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
