# Peer review of "Continuous online-monitoring of Ice Nucleating Particles: development of the automated Horizontal Ice Nucleation Chamber (HINC-Auto)"

_Atmospheric Measurement Techniques, 2020_

## Referee Comment (RC1) · Anonymous Referee #1 · 9 Oct 2020

Review of "Continuous online-monitoring of Ice Nucleating Particles: development of the automated Horizontal Ice Nucleation Chamber (HINC-Auto)" by Brunner and Kanji.

In this technical manuscript, the authors present a new instrument, HINC-Auto, allowing continuous measurements of INP concentrations at fixed temperature and humidity conditions. This instrument is based on the design of the HINC chamber with modifications brought in order to remove the need of a human operator. It is the first paper to report such instrument capable to measure INP concentration online in an automatic way for period that can extend several months. The scientific approach is well built

and does not raise any of my concerns. Furthermore, this manuscript is well written and easy to follow. This manuscript is a very valuable source of technicality for the ice community I am in favor of publication after the authors have answer the few questions and recommendations listed below:

Comments:

Line 18: "The interaction between aerosols and clouds contributes to the global energy budget by directly influencing the radiative forcing of the climate system." should read: "The interaction between aerosols and clouds contributes to the global energy budget by indirectly influencing the radiative forcing of the climate system."

line 66: Presently, no automated online INP counter is available (Cziczo et al., 2017; Lacher et al., 2017). A novel paper has just been submitted to AMTD, several weeks after the present one, showing another automatic online INP counter (expansion type chamber): https://amt.copernicus.org/preprints/amt-2020-307/

Line 92: "The surrounding sheath air is dried and filtered before entering the chamber. missing "dried"

Line 121: "A polyvinylidene fluoride (PVDF) spacer physically and thermally separates the two chamber walls."

Figure 3: Red and Green is a bad color combination as it is the most common form of colorblindness.

Figure 3a: How thick is the ice layer and is it included in the calculations (solid line)? The ice layer might decrease the actual volume of the chamber, increase the flow velocity and thus decrease the residence time of particles.

Figure 3: It is confusing. If I understood correctly, the message here is that "To smooth the flow field within the chamber and achieve a more consistent desired unidirectional flow field, a mesh was introduced 20 mm downstream of the sheath air injector holes." For clarity, maybe the legend of Figure 3 could be changed to: "Figure 3. Calculated

and measured particle residence time in a) HINC (without mesh) for different injector positions at T = 253 K and b) HINC-Auto (using a mesh to make the flow more laminar) at T= 243 K. Box plots from pulse experiments: median with 25/75% quartiles, whiskers: 5/95% quantiles. Median of PIV experiment (circles,T = 288 K) and CFD simulation (crosses, T = 243 K)." or maybe comparing HINC auto with and without mesh would be more relevant?

Figure 4:- "residence time until outlet" is not very intuitive for this figure. I would suggest replacing it by "residence time after entering the chamber" and reverse the time scale. In that way, it is simpler to compare both chamber. The main message here being the effect of cooler temperature entering the chamber, not the total length of the chamber. That way it is easier for the reader to read, that with cooler air, it needs only 20 cm and xx second to reach equilibrium, compare to 30 cm in yy second at warmer sheath air temperature. Also for a and b legend: I suggest to put temperature first as it is the reason, and length second as it is the consequence. a) T sheath air=298K, (original length, maybe not needed) b )Tsheath air=248 K, (10 cm shorter, maybe not needed)

line 210: "The chamber is controlled via a newly developed guided user interface programmed using Python 3.7 and corresponding open source packages. The postprocessing of INP concentrations is done in real time. HINC-Auto can be accessed and controlled remotely if a internet connection is available on site." A suggestion to the authors is to add in the appendix, a screen shot of the software interface could be presented, together with basic parameter that can be set by the user (RH ramping, Temperature profile,...).

line 233: "AF is the ratio of all particles, that are detected in the indicated size bin, to all sampled particles, measured with a CPC within the sample flow." A comment about the smaller particle: The authors discuss about the bigger size of aerosol that can enter HINC auto without being lost due to gravitational loss after activation within HINC-auto. However, what about the other extreme of the particle size distribution, the smaller particle? What is the (needed) detection limit of the CPC, and why this

detection limit was chosen? in other words, is there a minimum size for the particle to enter HINC that could have a chance to activate but not to grow enough to be detected as ice crystal? There is no restriction for small size particle to enter the chamber, would a CPC with very low cutoff size recommended? Or is there enough small particle loses due to diffusion prior to enter the chamber, in that case what is the cut of size of HINC? or in other word, how can we be sure that all particle that enter HINC and are counted in CPC, can have the possibility to grow big enough to be counted as ice crystal? Can HINC-auto able to measure INP properties of new particle formation that are of size of 5 nm?

Fig6a) Why the activated fraction is not 0 at the beginning? is it a real AF or a background of big non activated particle?

Fig6a) Why in opc channel >0.3 um there is a small but steady increase of the AF between 0.85 to 0.98 Sw? Is it due to the hygroscopic growth of particle just smaller than 300nm, which with higher humidity grow bigger than 300 nm?

line 261: "In either case, the temperature increases in the direction of the air flow, because of the parallel-flow setup of the cooling liquid (see Figure 2b). Therefore, the resulting total temperature variation in the center between the two cooling walls is T $\pm$ 0.24 K." I don't understand how the authors get this value of +/-0.24K.

line 270: "The CPC used for validation experiments has a counting uncertainty of $\pm$ 10% which yields in a relative uncertainty in the reported AF of $\pm$ 14%" same comment as earlier. how the detection threshold of the CPC affect the AF?

paragraphe 3.2.1: -So here the improvement is due to the flow which is more laminar in HINC-auto, correct? There is no mention of this in the paragraph. -Is this done at particle of size 200 nm? It is not mention in the text. And if it is monodisperse at 200 nm, why is there an increase of AF bellow 0.975 Sw?

Line 297: "The sheath to sample flow ratio was 12:1 for both chambers." It was stated

Line 267 that the sheath to sample flow ratio should be 9:1. Why is the ratio changed?

3.2.1 Improvement in precision: Why has −40C been chosen to show the homogeneous freezing onset? At this temperature, the water saturation is very close to the Koop line and it might be hard to distinguish between both. Going lower in temperature would show two distinct activation points.

Line 356: "The for the initial scan (solid line) the sheath flow set to 5 L min−1, and for the second scan to 2 L min−1 (dashed line)." please correct. Also, authors could state that at lower sheath flow, SMPS covers bigger particle size while losing smaller particle size.

line 359: "This is contradicting our assumption and the lower sheath flow rate of the DMA for the second compared to the ïñА̧rst measurement could be the reason." Could it be that with different sheath flow rates, the author can measure at different size range? And also during the time in the stainless steel chamber, coagulation of particle could have made the particles to grow bigger.

line 364: "For atmospheric relevant conditions at the JFJ with 400 cm−3 ≤ N ≤ 1000 cm−3 the drop occurs after 8.5 hours, for 95 cm−3 ≤ N ≤ 200 cm−3 after 13 hours." - how the authors define the drop? (drop of 20%?) -while the data agree with authors statement (data started at 25/12 and at almost 26/12), later in the experiment, after time 2, the data seems to suggest a need of more frequent rewetting. especially the experiment just after "time 2" and the next one.

line 369: "Based on the above experiment, we chose a rewetting time of 10 hours for field applications (Section 3.3)." As the rewetting is automatic and fast, why not choosing a more frequent rewetting to ensure good data, (i.e. to be on the safe side)? or did the author choose a 10 hours rewetting interval because at higher altitude, residence time is shorter, and in result the water depletion is longer?

line 378: "Design changes implemented in a second field campaign started in February

2020 resulted in a median LOD of 1.37 std L−1" Could the author specify which design change has been made, which allows a decrease by factor of almost 3 of the LOD? This would be very valuable for the INP community.

typo: caption fig 8: "with identical particle residence times. of" remove "."

---

## Referee Comment (RC2) · Anonymous Referee #2 · 17 Oct 2020

In this paper authors described the HINC-Auto, which is automated version of HINC. They describe the technical setup and validation experiments. CFD modeling is also performed. The chamber was deployed in the field and ran for 90 days. The paper is well written, and I recommend publication after following minor points are addressed.

Line 104 -105: Is flow rate affects the buoyancy? Can you have larger gradient but smaller flow rate?

Figure 2: For completeness label the vacuum pump. Currently, the output air is recirculated – closed loop configuration. What is the need of MFC after OPC?

Line 123: How many layers of filter paper were used.

Section 2.1: A plot showing the time series of temperature, RH, and droplet diameter would help to understand the operation of the chamber. This plot can include droplet growth, rewetting, and INP measurement periods.

Line 189: CDF or CFD?

Section 2.4: Regarding particle losses. Is there any size-dependent transmission curve established? Any particular reason why >3 um cannot be transmitted? Because of the presence of any upstream impactor?

How droplet diffusion growth calculations are performed? Can this equation be added to the appendix? It is not clear why d0 = 2 um used. A typical size-distribution at JFJ can be shown to understand what is d0.

How ice crystals are distinguished from water droplets? The description on line 238-240 is not clear. It is also mentioned that d = 0.2 um grew to 4.57 um. This indicates droplets and ice crystals co-exist. Please clarify.

Equation 1: Define the term [LOD] in the RHS or it is saying the units of LOD are in std Lmin-1 – if so move the units to another line. It is not clear how this equation is formulated. The number '60' in the numerator is confusing. Is this number not used to convert the 'V' into std L per sec from std L per min? If so then units of 'V' should be revised. What are the units of 'BG_counts' parameter?

Line 250: How many OPC intervals were used, and are they have the same length in terms of time? MDC = 1 count is defined. How this is assumed or calculated?

Line 269: clarify '. . .size bin 4995. . .'

Line 313: Is AF is same FF?

Figure 9: Add vertical line Sw = 1.04 to panel b to understand AF value. Please comment on AF. Do you achieve maximum droplet activation?

Figure 10: The tail end of the size distribution is not shown. It looks significant number of large particles exist. How these large particles (> 1 um) are distinguished from ice crystals?

Section 3.3: It is not very clear how AF/FF values are converted to std L min-1 as shown in Figure 11. Please show the equation. Do you use std Temperature and Pressure values?

It is not clear here, but how data is quality controlled? How data is flagged as good or bad. Any outliers are removed? Thoughts on data quality assessment would useful.

---

## Author Comment (AC2) · 23 Nov 2020

Reviewer comments are reproduced in **bold** and author responses in normal typeface; extracts from the original manuscript are presented in *red italic*, and from the revised manuscript in *blue italic*.

**In this technical manuscript, the authors present a new instrument, HINC-Auto, allowing continuous measurements of INP concentrations at fixed temperature and humidity conditions. This instrument is based on the design of the HINC chamber with modifications brought in order to remove the need of a human operator. It is the first paper to report such instrument capable to measure INP concentration online in an automatic way for period that can extend several months. The scientific approach is well built and does not raise any of my concerns. Furthermore, this manuscript is well written and easy to follow. This manuscript is a very valuable source of technicality for the ice community I am in favor of publication after the authors have answer the few questions and recommendations listed below:**

We would like to thank the reviewer for the compliments and valuable comments and address the comments individually below.

**Line 18: "The interaction between aerosols and clouds contributes to the global energy budget by directly influencing the radiative forcing of the climate system." should read:**
**"The interaction between aerosols and clouds contributes to the global energy budget by indirectly influencing the radiative forcing of the climate system."**

A critical error, thank you! We agree with the reviewer, and changed the sentence as proposed (see line 18 revised manuscript):

*The interaction between aerosols and clouds contributes to the global energy budget by indirectly influencing the radiative forcing of the climate system.*

**Line 66: Presently, no automated online INP counter is available (Cziczo et al., 2017; Lacher et al., 2017). A novel paper has just been submitted to AMTD, several weeks after the present one, showing another automatic online INP counter (expansion type chamber): https://amt.copernicus.org/preprints/amt-2020-307/**

We agree with the reviewer, and now include reference to this paper. We correspondingly altered Line 64 (revised manuscript) as stated below. In addition, we learned from the work by Bi et al., 2019, https://doi.org/10.1029/2019JD030609, in which they presented a continuous online INP counter.

*A prime limitation for the absence of long term monitoring data sets was that online real-time measurements of INP concentrations via INP counters required human operators as no autonomous online INP counter were available. Bi et al. (2019) presented the first autonomous online INP counter based on a CFDC. A novel paper by Möhler et al. (2020) introduced the Portable Ice Nucleation Experiment (PINE), an autonomous online INP counter that uses the adiabatic cooling during expansion to activate the INPs at the targeted supersaturation.*

**Line 92: "The surrounding sheath air is dried and filtered before entering the chamber. missing "dried"**

We agree with the reviewer, and changed line 94 (revised manuscript) as proposed:

*The surrounding sheath air is dried and filtered before entering the chamber.*

**Line 121: "A polyvinylidene fluoride (PVDF) spacer physically and thermally separates the two chamber walls."**

We agree with the reviewer, and changed line 123 (revised manuscript) as proposed:

*A polyvinylidene fluoride (PVDF) spacer physically and thermally separates the two chamber walls.*

**Figure 3: Red and Green is a bad color combination as it is the most common form of colorblindness.**

We agree with the reviewer, and changed Figure 3 such that color is no longer required to read it:

[Figure]

*Figure 3.*

**Figure 3a: How thick is the ice layer and is it included in the calculations (solid line)?**
**The ice layer might decrease the actual volume of the chamber, increase the flow velocity and thus decrease the residence time of particles.**

This is a valid statement. The filter paper is 0.5 mm think, if fully soaked with water and frozen 0.6 mm. This translates to a chamber height of 18.8 mm for a freshly replenished ice layers and 19 mm for depleted ice layers. The flow velocity of the center lamina increases by 1.1 % from 19.33 mm/s to 19.54 m/s. We attribute this change to be minor in light of the uncertainty in lamina thickness, $S_w$ and T of the chamber.

**Figure 3: It is confusing. If I understood correctly, the message here is that "To smooth the flow field within the chamber and achieve a more consistent desired unidirectional flow field, a mesh was introduced 20 mm downstream of the sheath air injector holes."**
**For clarity, maybe the legend of Figure 3 could be changed to: "Figure 3. Calculated and measured particle residence time in a) HINC (without mesh) for different injector positions at T = 253 K and b) HINC-Auto (using a mesh to make the flow more laminar) at T = 243 K. Box plots from pulse experiments: median with 25/75% quartiles, whiskers: 5/95% quantiles. Median of PIV experiment (circles, T = 288 K) and CFD simulation (crosses, T = 243 K)." or maybe comparing HINC auto with and without mesh would be more relevant?**

We agree with the reviewer, and changed the figure caption of Figure 3 as followed:

*Figure 3. Calculated and measured particle residence time in a) HINC (without mesh) for different injector positions at T = 253 K and b) HINC-Auto (using a mesh to achieve a more uniform flow) at T = 243 K. Box plots from pulse experiments: median with 25/75% quartiles, whiskers: 5/95% quantiles. Median of PIV experiment (circles, T = 288 K) and CFD simulation (crosses, T = 243 K).*

**Figure 4:- "residence time until outlet" is not very intuitive for this figure. I would suggest replacing it by "residence time after entering the chamber" and reverse the time scale. In that way, it is simpler to compare both chamber. The main message here being the effect of cooler temperature entering the chamber, not the total length of the chamber. That way it is easier for the reader to read, that with cooler air, it needs only 20 cm and xx second to reach equilibrium, compare to 30 cm in yy second at warmer sheath air temperature.**

**Also for a and b legend: I suggest to put temperature first as it is the reason, and length second as it is the consequence. a) T sheath air=298K, (original length, maybe not needed) b )Tsheath air=248 K, (10 cm shorter, maybe not needed)**

We agree with the reviewer, and changed the figure legend, time axis scale, and caption of Figure 4 as follows:

[Figure]

*Figure 4. Simulated ice supersaturation development along the chamber's center lamina at T = 243 K and contributing factors diffusion of heat and water vapor for a) $T_{sheath\ air}$ = 298 K (original length) and b) $T_{sheath\ air}$ = 248 K (allowing for the chamber length to be reduced by 10 cm).*

**Line 210: "The chamber is controlled via a newly developed guided user interface programmed using Python 3.7 and corresponding open source packages. The postprocessing of INP concentrations is done in real time. HINC-Auto can be accessed and controlled remotely if a internet connection is available on site." A suggestion to the authors is to add in the appendix, a screen shot of the software interface could be presented, together with basic parameter that can be set by the user (RH ramping, Temperature profile,. . .).**

We agree with the reviewer, and changed the paragraph as follows:

*The chamber is controlled via a newly developed guided user interface programmed using Python 3.7 and corresponding open source packages. The postprocessing of INP concentrations is done in real time. HINC-Auto can be accessed and controlled remotely if an internet connection is available on site, however this is*

*not a requirement for autonomous operation. A screenshot of the guided user interface with comments on the basic parameters a user can set is shown in the appendix (A7-A10).*

We further added the following subsection to the appendix:

[Figure]

*Figure A8. Main window of the guided user interface used to control HINC-Auto. Top row buttons: **Ramp Page** leads to the window to run $S_w$- or T-ramps (see Figure A9). **INP measurement** leads to the window to run INP measurements (see Figure A10) and **Settings Page** leads to the window to change additional settings (see Figure A11). **OPC MFC** shows the current flow rate of the OPC mass flow controller (MFC), as reported by the MFC, **Sheath MFC** analogously shows the flow rate of the sheath flow MFC, **Sample** reports the difference between both MFC, which corresponds to the current sample flow rate. **BGRamp** shows the target state of the chamber, in this case "measuring the background after or before a RH- or T-ramp, **BG** shows the state in which the chamber currently is, **rewetting at** indicates the time of the next planned rewetting procedure (UTC), **Sheath** indicates the RH and pressure p of the sheath air as it enters the chamber (measured at mid-height within the chamber upstream of the mesh, thus at the same temperature as the chamber is set to) and **Sample** indicates the RH and pressure p of the sample air (just downstream of the sample diffusion dryer. Center row buttons: **Set Twarm** sets the target temperature of the warm wall, **Set Tcold** sets the target temperature of the cold wall, **Set kp** and **Set ki** set the proportional gain ($k_p$) and integral gain ($k_i$) of the PI-controller to control the warm wall temperature, **Set valve loop time (s)** defines the duration of a loop to actuate the warm wall solenoid valve. Bottom row buttons: **open valve** manually opens and closes the warm wall solenoid valve, **Set display period (min)** defines the amount of historic data to be shown in the four graphs below, **turn drainage on** manually actuates the pump to drain the chamber during rewetting, **Rewet chamber** executes the rewetting procedure, **Rewet chamber and cool** executes the rewetting procedure and cools the chamber back down to the set temperature of the chamber before the button was activated, **Measuring** changes the chamber's state and valves to sampling or background measurement. The top left graph shows the warm wall temperature (all three thermocouples, the computed mean wall temperature and the set temperature). The top right graph shows the cold wall temperature (all three thermocouples, the computed mean wall temperature and the set temperature). The bottom left graph shows the counts reported by the OPC for each of the six size-bins/ channels (here CH 1 set to ≥0.3*

*μm, CH 2 set to ≥1 μm, CH 3 set to ≥3 μm, CH 4 set to ≥4 μm, CH 5 set to ≥5 μm, CH 6 set to ≥6 μm). The bottom right graph shows $S_w$ and T for the center lamina (calculated).*

[Figure]

*Figure A9. Window of the guided user interface used to control $S_w$ - or T-ramps with HINC-Auto.*

[Figure]

*Figure A10. Window of the guided user interface used to control INP measurements with HINC-Auto.*

*Figure A11. Window of the guided user interface used to adjust additional settings with HINC-Auto.*

**Line 233: "AF is the ratio of all particles, that are detected in the indicated size bin, to all sampled particles, measured with a CPC within the sample flow." A comment about the smaller particle: The authors discuss about the bigger size of aerosol that can enter HINC auto without being lost due to gravitational loss after activation within HINC-auto. However, what about the other extreme of the particle size distribution, the smaller particle?**

We like to thank the reviewer for this important point, since we did not consider commenting on the lower cut-off size of the particles entering the chamber.

**What is the (needed) detection limit of the CPC, and why this detection limit was chosen?**

The lower the cut-off size is of the CPC the more accurate the ambient aerosol particle counts will be. In the case of the experiments in this work, the CPC has a cut-off size of $D50 = 5$ nm. This was not actively chosen because for the laboratory measurements we performed experiments with known particles larger tens of nm. The CPC available at the JFJ has a higher size cut-off of $D50 = 10$ nm. This was also not chosen by us, but it is operated by a different institute under the GAW program.

**in other words, is there a minimum size for the particle to enter HINC that could have a chance to activate but not to grow enough to be detected as ice crystal?**

For the sampling conditions in HINC (T = 243 K, $S_w$ = 1.04, immersion mode), the smallest particle with the properties of NaCl to be activated would have an initial diameter of 11.4 nm according to Köhler theory. The Kelvin term quickly dominates Köhler theory for these small sizes, thus, a small decrease in initial radius leads to a larger needed increase in activation $S_w$. Depending on the chemistry of the particle, the initial radius may vary because of the van't Hoff factor or analogous for organic species. If we assume for the solution droplet to immediately freeze after activation it will grow to a diameter of 10.6 µm when having a residence time of 6 seconds in the chamber (diffusional growth by Rogers and Yau (1989)), large enough to be detected by the OPC and differentiated from droplets. We expect that particles with sizes < 12 nm will

therefore not activate into droplets and therefore not freeze, unless they activate by deposition nucleation. Furthermore, we argue that if such particles cannot activate as droplets and freeze by immersion at $S_w$ = 1.04, then they will also not be relevant for ice nucleation in the atmosphere. However, once they coagulate or grow by condensation to sizes larger ( > 20 nm) they could activate at lower $S_w$ (< 1.04), which tends closer to atmospherically relevant, at which stage such particles can also activate at our sampling conditions and can be sampled by HINC-Auto. Based on particle loss calculations (von der Weiden et al., 2009) 50% of particles with d = 10 nm and 98% of particles with d = 3 nm are lost in the tubing upstream of the chamber at JFJ.

**There is no restriction for small size particle to enter the chamber, would a CPC with very low cutoff size recommended?**
A CPC with a very low cutoff point is not necessary in this case because small particles (< 12 nm) would not activate in HINC-Auto, but also not activate in the atmosphere given that $S_w$ > 1.04 would be required as such, these do not need to be counted for estimating atmospherically relevant INP concentrations. Particles smaller than 12 nm resulting from new particle formation (NPF) for example would have to grow by coagulation to become of sizes large enough to activate into droplets and act as INPs in the immersion mode (or in the deposition mode at T < 235 K), however since INP activity scales with surface area, the contributions of particles below 12 nm is not of concern, in fact studies have proposed 100 nm to be the lower size limit for INPs (Pruppacher and Klett, 1997).

**Or is there enough small particle loses due to diffusion prior to enter the chamber, in that case what is the cut of size of HINC? or in other word, how can we be sure that all particle that enter HINC and are counted in CPC, can have the possibility to grow big enough to be counted as ice crystal? Can HINC-auto able to measure INP properties of new particle formation that are of size of 5 nm?**

In short no, 5 nm particles would need $S_w$ > 1.04 to activate as immersion INPs. So any chamber running at $S_w$ < 1.04 will not be able to account for INPs arising from a 5 nm particle due to Köhler theory limitations, regardless of diffusion losses. However, as we present above, the penetration size of particles into HINC-Auto is 10.3 nm ($d_{50}$) with 98% of 3 nm particles being lost (particle loss calculator, von der Weiden et al., 2009). Needless to say, all of these scenarios described by the reviewer above would require small particles to grow further to act as INPs even in the atmosphere, as such a 5 nm particle can also not act as an INP in the atmosphere, since it will not activate at atmospherically relevant $S_w$. So if particles of 5 nm (or less than 12 nm) do not activate in HINC, these will also not activate to droplets and eventually INPs in the atmosphere, unless they grow larger first by coagulation in which case they can be sampled by and activated in HINC-Auto. As such this is not of concern here.

In summary, the losses from diffusion of very small particles entering the tubing are deemed negligible for the measured INP concentration and thus need not be accounted for here. Further, diffusion losses in our laboratory experiments are also considered negligible because we focused on monodisperse particles of 200 nm or polydisperse particles of illite, where particle concentrations below 50 nm are already quite low and not ice-active (Welti et al., 2009) making it a negligible concern for particles below 12 nm.

**Fig6a) Why the activated fraction is not 0 at the beginning? is it a real AF or a background of big non activated particle?**

We assume the reviewer refers to the high AF in the 0.3 um channel. We believe that the observed ~15% particles above 0.3 μm are due to double (can be up to 25%*) and triple (up to 5.4%*) charged particles when size selecting 200 nm $NH_4NO_3$ particles (*calculations see Wiedensohler, 1988). A 5-minute background measurement is carried out before and after each ramp, thus at the low and high end of the tested saturation. The background counts are then linearly interpolated between the saturation and subtracted from the counts obtained during the measurement. The non-zero background count can additionally result from large particles that penetrate the DMA due to the transfer function used. In this

particular experiment, we used a lower sheath to sample flow, resulting in a broader transfer function, thus reducing the quality of mono-disperse selection. We now clarify this in the manuscript in line 326.

*The sample preparation is as described in section 3.2 for both chambers with a lower DMA sheath flow of 5 L min$^{-1}$ and a higher sample flow of 1.6 L min$^{-1}$ to feed both chambers and the CPC. This resulted in a broader transfer function within the DMA and consequently more larger and multiple charged particles penetrating the size selection. Due to the hygroscopicity of ammonium nitrate, the multiple charged particles are detected in the $\geq 1\ \mu m$ OPC size bin after hygroscopic growth at $S_w < 0.98$. In comparison, measurements in section 3.2 and Figure A5 use a narrower DMA transfer function and show a lower activated fraction below $S_w < 0.98$ than in the experiment with the broader transfer function. The injector position was chosen to result in residence times of $\tau \approx 9$ seconds.*

**Fig6a) Why in opc channel >0.3 um there is a small but steady increase of the AF between 0.85 to 0.98 Sw? Is it due to the hygroscopic growth of particle just smaller than 300nm, which with higher humidity grow bigger than 300 nm?**

The reviewer's statement is consistent with our understanding of the observed increase in AF. The fraction of size selected particles with a targeted mobility diameter of 200 nm (and double, triple charged particles) that grow because of hygroscopic growth to sizes with an optical diameter of 300 nm or larger increases steadily as the relative humidity is increased from $S_w = 0.85$ to 0.98.

**Line 261: "In either case, the temperature increases in the direction of the air flow, because of the parallel-flow setup of the cooling liquid (see Figure 2b). Therefore, the resulting total temperature variation in the center between the two cooling walls is T ± 0.24 K." I don't understand how the authors get this value of +/-0.24K.**

We like to thank the reviewer for pointing out this uncertainty. We aim to clarify by adding figure 7 (revised manuscript):

*3.1 Accuracy*

[Figure]

(1): -0.505 ± 0.14 K     (2): -0.095 ± 0.14 K     (3): 0.095 ± 0.14 K
(2+3): -0.095 - 0.14 to 0.095 + 0.14 K = ±0.24 K

*Figure 7. Schematic of uncertainty in temperature measurement showing a side view of HINC-Auto. TW and TC refers to one of six thermocouples installed on the warm wall and cold wall, respectively. Positions (1), (2), (3) and (2+3) indicate the temperature uncertainty of the center lamina. For each of the positions (1), (2), and (3) we have the ± 0.1 K form the warm and ± 0.1 K from the cold wall thermocouple, thus ± $\sqrt{0.1^2 + 0.1^2}$K = ± 0.14.*

*Four main parameters characterize the INP concentration measured: temperature, supersaturation, particle count and volume flow. The thermocouples have an uncertainty of ± 0.1 K and are calibrated measuring the melting of $H_2O$ and Hg, in close agreement with the ITS-90 (the official protocol of the international*

*temperature scale). The measured relative (compared to set point T) temperature variation across the warm and cold wall is -0.56/ +0.14 K and -0.45/ +0.05 K at T = 243 K and Sw = 1.04, respectively (see Figure 7). However, on each wall only the two thermocouples close to the injector (TW2/TC2) and the chamber outlet (TW3/TC3) are used to calculate the mean wall temperature. The relative variation therefore decreases to ±0.14 K (at the warm wall) and ±0.05 K (at the cold wall) for a center nominal temperature of T = 243 K at Sw = 1.04. In either case, the temperature increases in the direction of the air flow, because of the parallel-flow setup of the cooling liquid (see Figure 2b). Subsequently, the uncertainty of the center lamina is -0.095 K for the relative variation plus ±0.14 K for the thermocouples uncertainty at location (2) and 0.095 K ± 0.14 K at location (3). Therefore, the resulting total temperature variation in the section relevant for particle nucleation or activation and growth between the two cooling walls is T ±0.24 K. The ...*

**Line 270: "The CPC used for validation experiments has a counting uncertainty of ± 10% which yields in a relative uncertainty in the reported AF of ± 14%" same comment as earlier. how the detection threshold of the CPC affect the AF?**

During the validation experiments (subsection 3.2) the sample particles ($(NH_4)_2SO_4$, NaCl, $NH_4NO_3$) were size selected to a mobility diameter of $d_m$ = 200 nm and in the sedimentation study (Subsection 3.2.2, AgI) to $d_m$ = 100 nm. We believe this should render particles smaller than size cut-off of the CPC used (TSI 3787, D50 = 5 nm) to be negligible contributors to the total particle count. For the INP measurement (Subsection 3.2.3) the NX Illite particles where polydisperse and the SMPS retrieved particle size distribution (Figure 10a) shows more than 4 orders of magnitude lower particle concentrations smaller than 20 nm than between 300 and 500 nm which is substantially below the stated 14% uncertainty in AF. Finally, for the lower size cut-off contribution to field observations please see response to comment about Line 233 above.

**Paragraphe 3.2.1: -So here the improvement is due to the flow which is more laminar in HINC-auto, correct? There is no mention of this in the paragraph. -Is this done at particle of size 200 nm? It is not mention in the text.**

We agree with the reviewer, and changed the paragraph 3.2.1. as followed:

*Figure 8 shows the activation curves of ammonium nitrate size selected to a mobility diameter of $d_m$ = 200 nm and measured at T = 233 K with HINC-Auto compared to measurements performed with HINC. The sample preparation is as described in section 3.2 for both chambers with a lower DMA sheath flow of 5 L min$^{-1}$ and a higher sample flow of 1.6 L min$^{-1}$ to feed both chambers and the CPC. This resulted in a broader transfer function within the DMA and consequently more larger and multiple charged particles penetrating the size selection. Due to the hygroscopicity of ammonium nitrate, the multiple charged particles are detected in the ≥ 1 μm OPC size bin after hygroscopic growth at $S_w$ < 0.98. In comparison, measurements in section 3.2 and Figure A5 use a narrower DMA transfer function and show a lower activated fraction below $S_w$ < 0.98 than in the experiment with the broader transfer function. The injector position was chosen to result in residence times of τ ≈ 9 seconds. The standard sheath to sample flow ratio of HINC-Auto was adjusted to 12:1 to be equal to the ratio used in HINC in order to compare the performance of the two chambers (note that the standard sheath to sample flow ratio of HINC-Auto is 9:1. See section 2.2). HINC-Auto shows an improved precision compared to HINC. We attribute the improvement to the use of the mesh and, subsequently, the more uniform flow within HINC-Auto compared to HINC without the mesh. For measurements in the field with HINC, a defined supersaturation (e.g. $S_w$ = 1.04), temperature and OPC-size bin (e.g. ≥ 4 μm) is used to quantify INPs. Therefore, fluctuations in the activation precision of the ≥ 4 μm size bin can lead to uncertainties in INP concentrations. In the example of HINC, this is equivalent to more than one order of magnitude, thus an improved precision improves the quality of the INP measurements. In addition, particle sedimentation, as expected by theory (see Section below), is visible in the activation curves of HINC-Auto at Sw ≥ 1.02 (see Section 3.2.2).*

**And if it is monodisperse at 200 nm, why is there an increase of AF bellow 0.975 Sw?**

We thank the reviewer for the question. See response to reviewer comment to Fig6a) where we address this concern. Furthermore, we address it in line 326 (revised manuscript).

**Line 297: "The sheath to sample flow ratio was 12:1 for both chambers." It was stated Line 267 that the sheath to sample flow ratio should be 9:1. Why is the ratio changed?**

The sheath to sample flow ratio of HINC-Auto was adjusted to match the ratio used for the experiments in HINC. This to avoid potential biases when comparing the two chambers (a thinner center lamina results in a steeper activation since the variation in observed temperature and supersaturation is decreased). We added following sentence to Line 297

*The standard sheath to sample flow ratio of HINC-Auto was adjusted to 12:1 to be equal to the ratio used in HINC in order to compare the performance of the two chambers (Note that the standard sheath to sample flow ratio of HINC-Auto is 9:1. See section 2.2).*

**3.2.1 Improvement in precision: Why has −40C been chosen to show the homogeneous freezing onset? At this temperature, the water saturation is very close to the Koop line and it might be hard to distinguish between both. Going lower in temperature would show two distinct activation points.**

We agree with the reviewer, for the performance of homogeneous freezing onset, we refer readers to Figure 7 (and Figure A5), where we show experiments below 233 K. The key message of Section 3.2.1 and intended with Figure 8 is to demonstrate the precision increased with HINC-Auto compared to HINC when running multiple experiments.

**Line 356: "The for the initial scan (solid line) the sheath flow set to 5 L min−1, and for the second scan to 2 L min−1 (dashed line)." please correct. Also, authors could state that at lower sheath flow, SMPS covers bigger particle size while losing smaller particle size.**

We agree with the reviewer, and changed Line 356 as followed:

* For the initial scan (solid line) the sheath flow was set to 5 L min-1, and for the second scan to 2 L min-1 (dashed line). At a lower sheath flow rate, the SMPS scan shifts to cover a larger range of particle sizes while limiting the scanning range at smaller particle sizes.*

**Line 359: "This is contradicting our assumption and the lower sheath flow rate of the DMA for the second compared to the first measurement could be the reason." Could it be that with different sheath flow rates, the author can measure at different size range? And also during the time in the stainless steel chamber, coagulation of particle could have made the particles to grow bigger.**

We agree with the input of the reviewer about coagulation, and changed Line 359 as followed:

*This contradicts our initial assumption of large particles sedimenting more quickly than small particles, thus shifting size distribution towards smaller particle sizes with time. A reason could be due to particle coagulation in the stainless steel aerosol chamber, which causes the observed shift in size distribution towards larger particle sizes.*

**Line 364: "For atmospheric relevant conditions at the JFJ with 400 cm−3 ≤ N ≤ 1000 cm−3 the drop occurs after 8.5 hours, for 95 cm−3 ≤ N ≤ 200 cm−3 after 13 hours."**
**- how the authors define the drop? (drop of 20%?) -while the data agree with authors statement (data started at 25/12 and at almost 26/12), later in the experiment, after time 2, the data seems to suggest a need of more frequent rewetting. especially the experiment just after "time 2" and the next one.**

The reviewer is correct in asking how we define a drop. Since the drop is gradual, it is not possible to have a stringent cutoff (like 20 or 30%). However, motivated by the reviewer comment we agree that a more

frequent rewetting should be implemented. We now choose to be more conservative and use the drop observed after 8.5 hours as the re-wetting time. As such we now adjust the recommendation to 8 hours for re-wetting and have also adjusted this for the ongoing field measurements (see website). We changed the following sentence Line 369 (initial manuscript) to

*Based on the above experiment we choose a rewetting time of 8 hours for field applications in remote areas such as Jungfraujoch.*

**Line 369: "Based on the above experiment, we chose a rewetting time of 10 hours for field applications (Section 3.3)." As the rewetting is automatic and fast, why not choosing a more frequent rewetting to ensure good data, (i.e. to be on the safe side)?**

We agree with the reviewer. We initially choose to do the rewetting sequence less often because of two reasons:
1. The (mechanical/ thermal) stress on the chamber is largest during the rewetting procedure. Frequent rewetting reduces the lifetime of the components and could increase probability of breakdown.
2. If data is acquired over a long period of several months, analysis of the data with subsequent filtering is possible. Because of the large amount of data, ideally will not compromise the quality of the results drastically. The time between two rewetting procedures can then be adapted (which we now do and changed to 8 hours).

**Or did the author choose a 10 hours rewetting interval because at higher altitude, residence time is shorter, and in result the water depletion is longer?**

No, although we support the reviewer's statement, we did not think of this when choosing the rewetting interval. The rewetting time has now been changed to 8 hours.

**Line 378: "Design changes implemented in a second field campaign started in February 2020 resulted in a median LOD of 1.37 std L−1" Could the author specify which design change has been made, which allows a decrease by factor of almost 3 of the LOD? This would be very valuable for the INP community.**

We agree with the comment of the reviewer and changed Line 378 as followed:

*Design changes implemented in a second field campaign started in February 2020 resulted in a median LOD of 1.37 std L$^{-1}$. It was observed that ice crystals and frost deposited within the cavity of the chamber outlet. We assumed for supercooled liquid droplets, which make up the majority of the hydrometeor population exiting the chamber, to impact on the surfaces where the Swagelok fitting (to connect the OPC) is inserted into the PVDF frame. The change in inner diameter from the cavity within the PVDF frame ($d_i$ = 10.2 mm) to the fitting ($d_i$ = 3.3 mm) is like a step. The design changes included using a conical drill bit (20°) to smoothen the connection between the chamber outlet and the fitting. In addition, the Swagelok fitting is warmed with a 10 W heat pad during the rewetting procedure to support the evaporation of residual condensate or molten ice that does not drain due to gravity while the chamber is tilted.*

**Typo: caption fig 8: "with identical particle residence times. of" remove "."**

We changed the caption of Figure 8 as proposed by the reviewer. Thanks for catching that.

References

Bi, K., McMeeking, G. R., Ding, D. P., Levin, E. J., DeMott, P. J., Zhao, D. L., Wang, F., Liu, Q., Tian, P., Ma, X. C., Chen, Y. B., Huang, M. Y., Zhang, H. L., Gordon, T. D., and Chen, P.: Measurements of Ice Nucleating Particles

in Beijing, China, Journal of Geophysical Research: Atmospheres, 124, 8065–8075, https://doi.org/https://doi.org/10.1029/2019JD030609, 2019.

Kunert, A. T., Pöhlker, M. L., Tang, K., Krevert, C. S., Wieder, C., Speth, K. R., Hanson, L. E., Morris, C. E., Schmale, D. G., Pöschl, U., and Fröhlich-Nowoisky, J.: Macromolecular fungal ice nuclei in Fusarium: effects of physical and chemical processing, Biogeosciences, 16, 4647–4659, https://doi.org/https://doi.org/10.5194/bg-16-4647-2019, 2019.

Möhler, O., Adams, M., Lacher, L., Vogel, F., Nadolny, J., Ullrich, R., Boffo, C., Pfeuffer, T., Hobl, A., Weiß, M., Vepuri, H., Hiranuma, N., and Murray, B.: The portable ice nucleation experiment PINE: a new online instrument for laboratory studies and automated long-term field observations of ice-nucleating particles, Atmospheric Measurement Techniques Discussions, pp. 1–43, https://doi.org/https://doi.org/10.5194/amt-2020-307, in review, 2020.

Pruppacher, H. and Klett, J.: Microphysics of Clouds and Precipitation, Springer Netherlands, 2 edn., https://doi.org/https://doi.org/10.1007/978-0-306-48100-0, 1997.

Rogers, R. R. and Yau, M. K.: A short course in cloud physics / R. R. Rogers and M. K. Yau, Oxford, Pergamon Press. Rosenfeld, 3, illustrated, reprint edn., 1989.

Von der Weiden, S.-L., Drewnick, F., and Borrmann, S.: Particle Loss Calculator – a new software tool for the assessment of the performance of aerosol inlet systems, 2, 479–494, https://doi.org/https://doi.org/10.5194/amt-2-479-2009, 2009.

Welti, A., Lüönd, F., Stetzer, O., and Lohmann, U.: Influence of particle size on the ice nucleating ability of mineral dusts, Atmospheric Chemistry and Physics, 9, 6705–6715, https://doi.org/10.5194/acp-9-6705-2009, 2009.

---

## Author Comment (AC3) · 23 Nov 2020

Reviewer comments are reproduced in **bold** and author responses in normal typeface; extracts from the original manuscript are presented in *red italic*, and from the revised manuscript in *blue italic*.

**In this paper authors described the HINC-Auto, which is automated version of HINC. They describe the technical setup and validation experiments. CFD modeling is also performed. The chamber was deployed in the field and ran for 90 days. The paper is well written, and I recommend publication after following minor points are addressed.**

We would like to thank the reviewer for the compliments and valuable comments and address the concerns individually below.

**Line 104 -105: Is flow rate affects the buoyancy? Can you have larger gradient but smaller flow rate?**

From our understanding, a larger temperature gradient between the warm and cold wall would exacerbate the buoyancy at the warm wall. The difference in buoyancy will introduce shear within the fluid. A smaller flowrate will lead to less shear within the fluid. Thus, increasing the temperature gradient while decreasing the flow rate in order to maintain the same shear is probable. Exactly how the flow rate would influence the buoyancy would require complex 3D fluid dynamics simulations. However, we note that this aspect pertains to vertical chambers, which is not the topic of this paper.

**Figure 2: For completeness label the vacuum pump. Currently, the output air is recirculated – closed loop configuration. What is the need of MFC after OPC?**

We agree with the reviewer's comment and incorporated the proposition in Figure 2b:

**Figure 2.**

The MFC after the OPC is needed to maintain a chamber outlet flow rate of 2.83 std L min-1. Controlling both outlet and sheath flow rate allows to indirectly control the sample flow rate without the need of a MCF in the sample flow: chamber outlet flow rate = sheath flow rate + sample flow rate. This is stated in line 147: The sample air flow rate of 0.283 std L min-1 results from the difference of the volume flow exiting the chamber through the OPC and the sheath air directed into the chamber.

**Line 123: How many layers of filter paper were used.**

One layer of filter paper was used. We changed line 124 (revised manuscript) as follows: The inner metal walls are each covered with one layer of self-adhering borosilicate glass microfibre filter paper (PALL 66217, 1  $\mu$ m, 8x10") which is wetted with water and acts as reservoir for water vapor in order to create ice and/or water (super)saturation.

**Section 2.1: A plot showing the time series of temperature, RH, and droplet diameter would help to understand the operation of the chamber. This plot can include droplet growth, rewetting, and INP measurement periods.**

We thank the reviewer for the comment. We already include a time series of RH in Figure 4 and A4 and discuss droplet/ice crystal growth in section 2.4. Showing one time series with temperature, supersaturation and hydrometeor growth is somewhat arbitrary, as the plot substantially changes for each set temperature, supersaturation, ambient pressure and injector position. Section 2.1 provides just background information over CFDCs without having introduced any concepts like injector position or rewetting/INP measurement periods. Therefore, we think it is too premature to show Figure 4, A4 or an adaption including hydrometeor growth at this point i.e. in section 2.1.

**Line 189: CDF or CFD?**

We meant, CFD – thanks for catching that. We changed line 191 (revised manuscript) accordingly:

The 3D CFD simulation revealed for the supersaturation to need a substantial part of the chamber length to equilibrium to the set conditions.

**Section 2.4: Regarding particle losses. Is there any size-dependent transmission curve established? Any particular reason why >3 um cannot be transmitted? Because of the presence of any upstream impactor?**

We measured the particle counts using an OPC during a Saharan dust event at the JFJ on February 9th, 2020. During the dust event, also particles with an optical diameter of  $4.0 \,\mu$ m were present. A 15-minutes cumulative measurement was performed outside next to the total aerosol inlet, at the sample line, where normally the injector of HINC-Auto would be connected (i.e. after the dryer and valve), and at the outlet of the chamber, while both chamber walls were held at 20 °C. Table 1RC2 shows the transmission fraction of the ambient particles. No upstream impactor was used. We believe the horizontally oriented injector tubing upstream of HINC-Auto and injector within HINC-Auto allow for sedimentation, which limits their transmission rate.

|                | 1.0 μm | 1.5 μm | 2.0 µm | 3.0 µm | 4.0 μm |
|----------------|--------|--------|--------|--------|--------|
| Outside, next  |        |        |        |        |        |
| to Total Inlet | 100%   | 100%   | 100%   | 100%   | 100%   |
| After Dryer +  |        |        |        |        |        |
| Valve          | 72%    | 59%    | 40%    | 21%    | 14%    |
| After Chamber  |        |        |        |        |        |
| Walls at 20°C  | 69%    | 58%    | 33%    | 0%     | 0%     |

Table 1RC2. Transmission fraction of ambient particles during a Saharan Dust event at the JFJ on February 9th, 2020.

Furthermore, we do not believe this transmission efficiency affects our results in a significant manner since we reach an AF = 0.98 for sampling ambient particles at JFJ (see figure 6, revised manuscript).

How droplet diffusion growth calculations are performed? Can this equation be added to the appendix? It is not clear why d0 = 2 um used. A typical size-distribution at JFJ can be shown to understand what is d0.

We thank the reviewer for highlighting the missing reference. The diffusional growth calculations are according to Rogers and Yau (1989). We added the citation to line 226 (revised manuscript):

*Diffusional growth calculations (Rogers and Yau, 1989) ...* (See continuation of text after the next **reviewer comment**)

Furthermore, we added the following passage to the appendix; line 545 (revised manuscript):

Diffusional growth is calculated according to Rogers and Yau (1989) with the latent heat of sublimation of ice, and latent heat of vaporization of supercooled water according to Murphy and Koop (2005). T and S are variable and feed in from the 2D diffusion model corresponding to the particle's current horizontal and vertical position within the chamber. The diffusional growth of the hydrometeors assumes activation when saturation with respect to ice or water is exceeded.

 $d_0 = 2\mu m$  was used, as it is the largest observed particle diameter able to transmit through the tubing and HINC-Auto at the JFJ. We changed line 221 (revised manuscript) as follows:

The transmission fraction of ambient particles  $\geq 2 \ \mu m$  through the tubing and the dry chamber (both walls held at 293 K) on the JFJ is 33%. No ambient particles  $\geq 3 \ \mu m$  were transmitted. Therefore, to assess the maximum size of droplets in the following diffusional growth calculations a maximum initial radius of  $d_0 = 2 \ \mu m$  is used.

How ice crystals are distinguished from water droplets? The description on line 238-240 is not clear. It is also mentioned that d = 0.2 um grew to 4.57 um. This indicates droplets and ice crystals co-exist. Please clarify.

This is a valid comment, for which we like to thank the reviewer. Stating first the maximum diameters assuming a constant T and  $S_w$ , then saying, that this is not how particles experience T and  $S_w$  within the chamber, as T and  $S_w$  need to equilibrate, and then stating the maximum diameters with variable, equilibrating T and  $S_w$  is confusing for the reader. Therefore, we altered lines 224-249 (revised manuscript) to explain this more clearly for constant T and  $S_w$ , and then quantitatively stating the maximum diameters with variable, equilibrating T and  $S_w$ :

Diffusional growth calculations (Rogers and Yau, 1989) with set fixed T (e.g. constant at 243 K) and  $S_w$  (e.g. constant at 1.04) conditions overestimate the final hydrometeor size at the chamber exit since the calculation assumes a constant supersaturation to be maintained for the entire time the particle passes through the chamber. In reality, the saturation in the particle stream needs to equilibrate to the set conditions, thus, the particles are exposed to a lower saturation for the first few seconds (see Figure A4). The 2D diffusion model provides an estimate of the real T and  $S_w$  when using the diffusional growth calculations by Rogers and Yau (1989). For an initial diameter of  $d_0 = 2 \mu m$ , liquid droplets are calculated to grow to a maximum size of  $d_{lia}$  = 3.31 µm (Zurich, 965 hPa,  $\tau$  = 9.1 s) and  $d_{lia}$  =2.36 µm (JFJ, 645 hPa,  $\tau$  = 6.1 s). Measurements of a highly hygroscopic aerosol, ammonium nitrate with an initial mobility diameter of  $d_m =$ 200 nm (for the sample preparation see Section 3.2) show the onset of cloud droplets (no ice crystals since T > 235 K) in the  $\geq$  3  $\mu$ m-size bin at Sw = 1.038, as seen in Figure 6a, and support the calculated maximum size of 3.31  $\mu$ m at Sw = 1.04. The impact on the final diameter for an initial size of  $d_{0.0,2}$  = 200 nm compared to  $d_{02.0} = 2 \mu m$  is 0.63  $\mu m$  ( $d_{lia2.0} = 3.31 \mu m$  vs.  $d_{lia0.2} = 2.68 \mu m$  at 965 hPa and  $\tau = 9.1$  s). If the INPs activate as soon as ice saturation is exceeded, the ice crystals grow to  $d_{ice 2.0} = 7.77 \mu m$ ,  $d_{ice 0.2} = 7.51 \mu m$  at 965 hPa and  $\tau = 9.1s$  and  $d_{ice_{2,0}} = 7.66 \,\mu m$  (JFJ, 645 hPa,  $\tau = 6.1 \,s$ ). Therefore, for experiments performed at T = 243 K and Sw = 1.04, all particles detected in the size bin  $\ge 4 \mu m$  are considered to be ice crystals formed on INPs. Figure 6b shows a measured activated fraction (AF) curve of ambient air on the JFJ during a high INP concentration period (7:05 22. March 2020, UTC). AF is the ratio of all particles, that are detected in the indicated size bin, to all sampled particles, measured with a CPC within the sample flow. The onset of cloud droplets in the  $\geq$  0.3  $\mu$ m size bin exactly at Sw = 1 demonstrates the accuracy of HINC-Auto. At Sw = 1.13 an observed steep increase in AF in the  $\geq$  3  $\mu$ m-OPC size bin indicates droplets only grew larger than 3  $\mu$ m at

this Sw. Compared to the ammonium nitrate measurements performed at Zurich, a delayed activation is observed. This is expected because of the decrease in ambient pressure, which results in shorter residence times, and the much lower hygroscopicity of ambient particles at the JFJ compared to ammonium nitrate. The signal visible in the  $\ge 4 \mu$ m-OPC size bin comes from INPs, which nucleate and grow to ice crystals at  $S_w \ge 1.028$  (Si  $\ge 1.378$ ). This validates the calculations above that at  $S_w = 1.04$  droplets cannot grow to sizes  $\ge 4 \mu$ m but ice crystals can, thus supporting the use of the  $\ge 4 \mu$ m size bin to detect ice crystals.

**Equation 1: Define the term [LOD] in the RHS or it is saying the units of LOD are in std Lmin-1 – if so move the units to another line. It is not clear how this equation is formulated. The number '60' in the numerator is confusing. Is this number not used to convert the 'V' into std L per sec from std L per min? If so then units of 'V' should be revised. What are the units of 'BG\_counts' parameter?**

The RHS term refers to the units of the LOD. We deleted the expression  $[LOD] = std L min^{-1}$ . Supported by the reviewer comment about **Section 3.3** (see the comment later on) we now show how the INP concentration is calculated. Therefore, some variables ( $\Sigma$ BG counts,  $\Sigma$ NBG samples, V and tOPC) are introduced there and not following the calculation of the LOD in equation (2). We changed line 256 (revised manuscript) as following:

**The INP concentration is calculated as follows:**

 $INP \ concentration = \left(\frac{\sum INP \ counts}{\sum N_{INP \ samples}} - \frac{\sum BG \ counts}{\sum N_{BG \ samples}}\right) \frac{1}{Vt_{OPC}}$ (1)

where  $\Sigma$ INP counts is the sum of all counts (particle number) in the  $\ge 4 \mu m$  OPC size bin during the INP measurement,  $\Sigma N_{INP \ samples}$  is the total number of OPC intervals during the INP measurement,  $\Sigma BG$  counts is the sum of the background counts (particle number) in the  $\ge 4 \mu m$  OPC size bin before and after the INP measurement while sampling through a particle filter,  $\Sigma N_{BG \ samples}$  is the total number of background OPC intervals before and after the INP measurement,  $t_{OPC}$  is the duration of each OPC interval in minutes (here 5 sec, thus 5/60 min), and V is the sample flow rate, here V = 0.283 std L min-1. As the volume flow through the OPC is controlled by the MFC in std L min-1, the resulting INP concentration is INP std L-1.

The limit of detection (LOD) is calculated as follows:

$$LOD = \frac{\sqrt{\sum BG \ counts}}{\sum N_{BG \ samples}} \frac{601}{V t_{OPC}} \qquad [LOD] = \text{std L min}^{-1} \tag{1} (2)$$

where the LOD is in std L-1. If over a period of 120 OPC background sampling intervals with a duration of 5 seconds each a total of 3 counts where detected in the  $\ge 4 \mu m$  OPC size bin, the LOD would be = 0.612 std L min-1 ( $\Sigma N_{BG \ samples} = 120$ ,  $t_{OPC} = 0.083 \text{ min}$ ,  $\Sigma BG$  counts = 3,  $V = 0.283 \text{ std L min}^{-1}$ ). The stated LOD provides a 62.3% (1  $\sigma$ ) confidence interval.

Concerning the number '60': it was used to convert the OPC duration from seconds to minutes. We changed Equation 1 by removing the '60' and changing the text in line 262 (revised manuscript) from seconds to minutes.

'BG\_counts' are just particles where each count represents a particle, so the unit could be particles or number (#). To clarify, we added *"(particle number)"* where we defined background counts in the OPC after equation 1. See line 258 (revised manuscript).

Furthermore, we realized we stated that V is the flow rate through the OPC, which is false. It should be the sample flow rate. We altered the passage in line 262 (revised manuscript) to

...and V is the sample flow rate, here V = 0.283 std L min-1.

**Line 250: How many OPC intervals were used, and are they have the same length in terms of time? MDC = 1 count is defined. How this is assumed or calculated?**

Each OPC interval is 5 sec in duration (can be set in the OPC). A typical background measurement is 5 minutes, thus 60 intervals. For every INP measurement, there is one background measurement before and after, so 120 intervals in total. The INP measurement is typically 15 min in duration, which is equivalent to 180 intervals.

The minimum detectable concentration is the smallest non-zero signal from the OPC, which is 1 count, divided by the volume of air sampled during the INP measurement, which is 15 min \* 0.283 std L min-1 = 4.245 std L. Therefore, for the defined INP sampling conditions the MDC is 1/4.245 = 0.2356 INP std L-1. To reduce the level of ambiguity we changed line 270 (revised manuscript) as follows:

The minimum detectable concentration (MDC) is 1 count (particle) in the  $\ge 4 \mu m$  OPC size bin over a 15minute INP measurement with a sample flow rate of 0.283 std L min-1 over 15 minutes, which equals MDC = 0.236 std L-1.

**Line 269: clarify '...size bin 4995...'**

According to the manufacturer, the used OPC (MetOne GT-526S) can count 4995 particles per second, and simultaneously classify their optical size and place them in one of 6 user-defined size bins. We changed line 298 (revised manuscript) accordingly:

According to the manufacturer, the used OPC can count 4995 particles per second, and simultaneously classify their optical size and place them in one of 6 user-defined size bins, with an overall accuracy of  $\pm$  10% to the calibrated aerosol.

**Line 313: Is AF is same FF?**

Yes, AF and FF are the same. We changed Figure X (revised manuscript) correspondingly: